



# Ocean Deoxygenation and Copepods: Coping with Oxygen Minimum Zone Variability

Karen F. Wishner[1], Brad Seibel[2], Dawn Outram[1]

[1]Graduate School of Oceanography, University of Rhode Island, Narragansett, RI 02882, USA
[2]College of Marine Science, University of South Florida, St. Petersburg, FL 33701, USA

*Correspondence to*: Karen Wishner (kwishner@uri.edu)

**Abstract.** Increasing deoxygenation (loss of oxygen) of the ocean, including expansion of oxygen minimum zones (OMZs), is a potentially important consequence of global warming. We examined present day variability of vertical distributions of copepod species in the Eastern Tropical North Pacific (ETNP) living in locations with different water column oxygen profiles and OMZ intensity (lowest oxygen concentration and its vertical extent in a profile). Copepods and hydrographic data were collected in vertically-stratified day and night MOCNESS tows (0-1000 m) during 4 cruises over a decade (2007-2017) that sampled 4 ETNP locations: Costa Rica Dome, Tehuantepec Bowl, and 2 oceanic sites further north (21°-22°N) off Mexico. The sites had different vertical oxygen profiles: some with a shallow mixed layer, abrupt thermocline, and extensive very low oxygen OMZ core, and others with a more gradual vertical development of the OMZ (broad mixed layer and upper oxycline zone) and a less extensive OMZ core where oxygen was not as low. Copepod species (including examples from the genera *Eucalanus*, *Pleuromamma*, and *Lucicutia*) demonstrated different distributional strategies and physiologies associated with this variability. We identified sets of species that (1) changed their vertical distributions and depth of maximum abundance associated with the depth and intensity of the OMZ and its oxycline inflection points, (2) shifted their depth of diapause, (3) adjusted their diel vertical migration, especially the nighttime upper depth, or (4) expanded or contracted their depth range within the mixed layer and upper part of the thermocline in association with the thickness of the aerobic epipelagic zone (habitat compression concept).

The upper ocean to mesopelagic depth range encompasses a complex interwoven ecosystem characterized by intricate relationships among its inhabitants and their environment. It is a critically important zone for oceanic biogeochemical and export processes and hosts key food web components for commercial fisheries. Among the zooplankton, there will likely be winners and losers with increasing ocean deoxygenation as species cope with environmental change. Changes in individual copepod species abundances, vertical distributions, and life history strategies may create potential perturbations to these intricate food webs and processes. Present day variability provides a window into future scenarios and potential effects of deoxygenation.

## 1 Introduction

Increasing ocean deoxygenation (loss of oxygen), including expansion of oxygen minimum zones (OMZs), is a potentially important consequence of global warming (Breitburg et al., 2018; Levin, 2018; Stramma et al., 2008). In the future, OMZs (oceanic depth zones of low oxygen) are predicted to expand in geographic extent and to have even lower oxygen concentrations





in their centers compared to the present. The zooplankton community from the mixed layer into the mesopelagic is a key component of oceanic food webs, particle processing, and biogeochemical cycles (Robinson et al., 2010; Steinberg and Landry, 2017). From earlier work, some mesopelagic species are known to be sensitive to environmental oxygen concentration and respond to that variability by altering their distribution patterns (Wishner et al., 2018, 2013). If the OMZ expands on a broader

scale, distributions and abundances of many zooplankton species will likely be affected, with potential consequences for oceanic ecosystems.

Here, we examine the variability of vertical distributions of copepod species living in locations with different water column oxygen profiles and OMZ intensity (lowest oxygen concentration and its vertical extent in a profile). We obtained quantitative

vertically-stratified zooplankton samples from the surface to 1000 or 1200 m, with simultaneous hydrographic data, from 4 cruises from 2007 – 2017 at several sites in the Eastern Tropical North Pacific (ETNP), an oceanic region with a strong OMZ (Fiedler and Talley, 2006; Karstensen et al., 2008; Paulmier and Ruiz-Pino, 2009). The tows sampled 4 different geographic locations, each showing a distinctive shape of the vertical oxygen profile through the OMZ. Some sites had a narrow shallow mixed layer, abrupt thermocline, and extensive very low oxygen OMZ core, while others had a more gradual vertical

development of the OMZ (broad upper oxycline zone of diminishing oxygen) and a less extensive OMZ core where the concentration of oxygen was not as low.

We previously noted the presence of zooplankton boundary layers (sharp peaks of high zooplankton biomass and abundances of certain fish and copepod indicator species) at the upper and lower edges (oxyclines) of the OMZ core at several of these sites

(Costa Rica Dome, Tehuantepec Bowl), and how the lower oxycline biomass peak shifted depth by tracking oxygen concentration (Wishner et al., 2013). These peaks of zooplankton concentration in the water column were also locations of active trophic processing (Williams et al., 2014), life history events (diapause depths) for some species (Wishner et al., 2013), and predator concentrations (Maas et al., 2014). On the 2017 cruise, horizontally-sequenced transects and tows at three different depths within the OMZ exhibited considerable submesoscale physical and biological variability (including km-scale features of

oxygen); zooplankton abundance differences were associated with oxygen differences of only a few μM (Wishner et al., 2018). Here, we focus on copepod vertical distributions in order to compare species-specific responses to large scale spatial and temporal environmental variability among the four different OMZ locations.

Many marine taxa, including zooplankton, are physiologically sensitive to the oxygen partial pressure in their environment

(Childress and Seibel, 1998; Seibel, 2011; Seibel et al., 2016). One measure, typically viewed as an index of hypoxia tolerance, is the lowest experimentally determined partial pressure of oxygen at which an individual of a species can maintain its metabolism ($P_{crit}$). Recent work demonstrates that hypoxia does not specifically select for a lower $P_{crit}$, but rather that $P_{crit}$ is a reflection of the physiological capacity to extract and transport oxygen in support of maximum activity (Seibel and Deutsch, 2019). Species living in extreme hypoxia, such as those in OMZs, have greater oxygen supply capacity for a given metabolic

rate than do species living in air-saturated water. As a result, measured $P_{crit}$s are generally related to the environmental parameters (temperature, oxygen, depth) where the species occurs (Childress, 1975; Seibel, 2011) and are measured much more commonly than is maximum activity. Mesopelagic species living in OMZs have the lowest measured $P_{crit}$s among animals (Seibel et al., 2016; Thuesen et al., 1998; Wishner et al., 2018). For most of these species, hypoxia tolerance is reduced at higher temperatures ($P_{crit}$s are higher) because respiration increases, but one copepod species (*Lucicutia hulsemannae*), living at the

OMZ lower oxycline, has a reverse temperature response, which is adaptive in this habitat where oxygen and temperature are





inversely correlated (Wishner et al., 2018). *In situ* abundances of some OMZ species responded to slight environmental oxygen variations at very low oxygen concentrations (<10 μM) at mesopelagic depths, corresponding to their experimentally determined physiological tolerances (Wishner et al., 2018). Here, we document oxygen and temperature ranges at the depth of maximum abundance in vertical profiles for a number of copepod species, as an indicator of the most metabolically suitable habitat for each
species at the time and place of collection.

Alterations in zooplankton distributions in response to oxygen may have wider implications for ecosystem function and ultimately drive potential societal impacts (e.g. fisheries effects). Present day comparisons among locations with different water column oxygen structure can be used as proxies to illuminate potential future effects of OMZ expansion. We analyzed
zooplankton vertical distributions spatially and temporally, using key copepod species as examples, to address the following important questions about OMZ expansion effects:

1. Do mesopelagic copepod species adjust their depth distribution, especially the occurrence of boundary layers at oxyclines, in association with the shape of the vertical profile of oxygen?

2. Does the depth of diel vertical migration (DVM) for particular species change in association with OMZ thickness and oxygen concentration, or do the copepods continue to migrate to the same depth regardless of oxygen?

3. Can distributional effects of aerobic habitat compression on epipelagic copepod species be documented? Specifically, do
mixed layer species have different lower depth boundaries depending on the depth and abruptness at which very low oxygen occurs?

## 2 Methods

### 2.1 Sampling

Zooplankton samples were collected in day and night vertically-stratified net tows during 4 cruises to the ETNP (Fig. 1, Table 1).
In Oct – Nov 2007 on *R/V Seward Johnson* and Dec 2008 on *R/V Knorr* (cruise 19502), sites at the Costa Rica Dome (CRD) (9°N, 90°W) and Tehuantepec Bowl (TB) (13°N, 105°W) were sampled as part of the ETP Project. A few years later as part of the Metabolic Index Project, *R/V Oceanus* (cruise OC1604B) in Apr 2016 occupied a station (OC) farther north at 22°N, 120°W, and the next year in Jan-Feb 2017, *R/V Sikuliaq* (cruise SKQ201701S) occupied a station (SKQ) at 21°N, 117°W.

At all locations, a series of vertically stratified MOCNESS (Multiple Opening Closing Net and Environmental Sensing System (Wiebe et al., 1985) tows, along with simultaneous hydrographic data, collected day and night zooplankton samples from depth (1000 or 1200 m) to the surface (Table 1, Table S1 (Wishner and Outram, subm.)). Standard tows sampled the entire 0-1000 m depth range in 100 or 200 m intervals (8 or 9 nets). At most stations, additional fine scale tows were taken over various intermediate depth ranges, usually at sampling intervals of 25, 50, or 150 m. Adaptive sampling targeting particular oxygen
values ("physiology" and "oxycline" tows) was also done. At SKQ, 2 sequential day tows (Tows 718 and 719) and 1 night tow (Tow 720), that targeted the 350-650 m depth zone with 8 sampling nets for each tow, opportunistically encountered different vertical oxygen profiles, and some species distributions responded to this variability. This tow series is referred to as the "mid-depth comparison". Results from horizontally-sequenced tows at specific mesopelagic depths within the OMZ, performed during the *R/V Sikuliaq* cruise, were previously published (Wishner et al., 2018).






Zooplankton were sampled on the upcast portion of a tow close to noon or midnight, and the hydrographic data from the MOCNESS sensors taken simultaneously with the sampling was used.  In a few instances when sensors failed, hydrographic information for some depths was used from the downcast portion of the tow or from a nearby CTD cast.  The MOCNESS was towed at a speed of ~1.5 kt along an oblique path; vertical speeds during the deeper sampling upcast were typically 1 to 3 m

min$^{-1}$.  Because OMZ zooplankton are sparse, large volumes of water (typically 600 to >1000 m$^3$) were filtered by each net for the deeper samples, and those nets were open for ~20 minutes each.  The duration of the deeper tows was ~7 h (most of a day or night, centered around noon or midnight).  Shallower tows were shorter in duration, and smaller volumes were filtered for the mixed layer and upper water samples.  Details of tows, depth intervals, and volume filtered for each net are in Table S1 (Wishner and Outram, subm.)**.**


By combining data from multiple tows for a given location to construct a full depth series, we were able to resolve fine scale vertical distributions within the OMZ and at its upper and lower boundaries.  The process for assembling vertical profiles from multiple tows was described in Wishner et al., (2013) for the 2007 and 2008 cruises; a similar process was used for the other cruises although depth strata varied (Table 1).

**2.2  MOCNESS instrument**

Two versions of the 1 m$^2$ MOCNESS were used.  The *R/V Seward Johnson* and *R/V Knorr* cruises used the University of Miami's classic MOCNESS with the standard MOCNESS deck unit, software, and sensors, with the oxygen sensor incorporated into the options unit.  The *R/V Oceanus* and *R/V Sikuliaq* cruises used the upgraded Scripps Institution of Oceanography's MOCNESS in which a Seabird 911CTD with hydrographic sensors and software was physically attached to the MOCNESS

frame and its software integrated with the MOCNESS system (Wishner et al., 2018); the bottle-closing CTD feature was transformed into a net bar release controller.  The basic net frame and nets, net bar release mechanism, and flow meter were the same in both systems.  The recent upgrades provided more stable hydrographic sensor data and facilitated real-time adaptive sampling with an improved shipboard data and graphical user interface.  In the two earlier cruises, the net mesh size was 153 μm; in the two later cruises it was 222 μm.  Sensors included pressure (depth), temperature, salinity, *in situ* fluorescence, light transmission, PAR (later cruises), and a TSK flow meter (for volume filtered).  The oxygen sensor for all cruises was a Seabird

SBE43, calibrated in advance of the cruises.

**2.3  Sample processing, copepod sorting, and abundance data**

At sea, cod ends upon retrieval were placed in buckets with plastic ice packs to maintain cool temperature in the tropical heat.  Fresh samples were rinsed with filtered seawater into large 153 μm mesh metal sieves and photographed.  Most samples from the

earlier cruises were then split in a flat-bottomed Motoda splitter.  Typically, a half split was preserved in 4% sodium-borate buffered formaldehyde for later species identification, while other splits were used for experimental studies, at-sea size-fractionated biomass measurements (first two cruises), photography, and stable isotope analyses.  Samples from the later cruises were usually preserved whole without splitting, except for a few individuals that were set aside for live experiments (and recorded at that time).


Onshore processing, including copepod sorting and identification, varied among the cruises because of time and funding constraints.  Onshore sorting of copepods occurred months to years after a cruise.  The 2007 CRD tows, most of the 2016 OC





tows, and most of the 2017 SKQ tows were sorted to substantial species-level and lifestage detail, while others (especially some 2007 and 2008 TB tows) were sorted for only a few selected species. For the 2016 and 2017 cruises, only larger copepods were

counted after the sample was poured through a 2 mm sieve in the lab, while copepods of all sizes were counted in the 2007 CRD samples (targeting major groups). Splits of the whole sample were used in most cases, with a target sample size of 100 calanoid copepods. In some instances for SKQ horizontal tows, Stempel pipette aliquots of the small or large size fractions were checked and quantified for particular taxa. About 22,000 identified copepods are included in this paper's dataset.

Certain copepod species were consistently identified and visually searched for in most tows, with a focus on the genera *Lucicutia* and *Pleuromamma* and the family Eucalanidae. Many other copepod species were identified when present, except for deeper TB samples (below 200 m) for which only targeted species were quantified. Some species, especially ones that were rarer, smaller, immature, or difficult to identify, were not always separated and were often lumped into broader categories such as genus, family, or "miscellaneous calanoid". Because adult morphological features are typically needed for species confirmation,

identification targeted adult calanoid copepods. For some species in which particular younger stages were abundant, easily recognizable, and essential for ecological understanding, those stages were counted separately. This was especially important for the copepod *Eucalanus inermis* which had an obvious diapausing layer sometimes associated with OMZ oxyclines. Copepods were identified by an experienced technician (Dawn Outram) based on the literature and websites, especially (Razouls, C. et al., 2005-2019), website http://copepodes.obs-banyuls.fr/en). Identifications of individuals were done by hand with Nikon and Wild

stereo and compound microscopes, enabling more detailed species and lifestage delineation than possible with bulk sample electronic scanning. An upgraded microscope with greater acuity was used for the 2016 and 2017 sample processing, allowing greater species resolution. Other sample components, including size-fractionated biomass, zooplankton stable isotope and body composition, and abundances of other taxa including euphausiids, fish larvae, cephalopods, and foraminifera are reported elsewhere or are currently under analysis (Birk et al., 2019; Cass et al., 2014; Cass and Daly, 2014, 2015; Maas et al., 2014;

Williams et al., 2014; Wishner et al., 2018, 2013).

Abundances within each net, usually reported as number of individuals $(1000 \text{ m})^{-3}$, were calculated after accounting for split size and volume filtered (Table S1 (Wishner and Outram, subm.)). Abundances in vertical distribution graphs are plotted at the mid-depth of the sampling interval. Water column abundance (reported as number of individuals $\text{m}^{-2}$ for a selected depth range,

(usually 0-1000 m) was calculated by multiplying each sample abundance by the depth range of that sample and then summing those values for the complete sample series over the total selected depth range.

### 2.4 Species selection for this paper and depth of maximum abundance

To clarify critical issues about OMZ effects on zooplankton, we selected a subset of 23 species from the extensive full species list (over 300 species) in order to highlight specific responses to OMZ variability (Table 2). We chose these species based on

their abundance and frequency of occurrence (presence/absence) in the different locations and cruises and their usefulness as examples to illustrate different types of distributional responses to variability in the shape (and lowest oxygen concentration) of the oxygen profile. In a set of graphs, we show day and night vertical distributions of particular species relative to the oxygen profile from each sampling location and time (when that species was present).

The depth of maximum abundance (DMA) was defined as the single net and its depth interval with the highest abundance for a particular species at each station during each cruise, with all tows considered (Table 3). The MOCNESS sensor data during the



time of collection (upcast) was used to delineate the habitat parameters (oxygen and temperature ranges) at the DMA; these are from the particular tow and net listed. Most species also occurred (at lower abundance) over a broader vertical range beyond the single DMA net, as evident in the figures and Table S1 (Wishner and Outram, subm.). Some species exhibited bimodal

distributions. It should be noted that because each net in a MOCNESS tow encompassed a range of depths and consequently a range of hydrographic values, it is unknown where within that range the collected animals were actually located. However, our fine scale sampling in many cases encompassed very narrow environmental intervals such that we could constrain the OMZ habitat preferences and tolerances (oxygen and temperature ranges) for many species.

## 3 Results

### 3.1 Environmental structure

Hydrographic profiles from MOCNESS sensors are shown in Fig. 2 for each location for both the full sampling range (0-1000 m) and the upper water column (0-200 m). We selected one oxygen profile from each site and year to represent that location in the vertical abundance graphs, although abundance profiles were derived from combining multiple day and night tows (see sect. 2.1 and 2.3). In a few cases, we discuss small scale distributional variability between sequential tows at the same location,

especially the "mid-depth comparison" series at SKQ.

The OMZ water column consists of several ecological zones, previously described, each of which has a characteristic hydrographic structure and forms the habitat for a suite of organisms and processes (Wishner et al., 2008, 2013). Zonal boundaries depend on both the shape of the oxygen profile and oxygen concentration, and boundary depths vary by profile. The

mixed layer is the water column above the thermocline with high oxygen and temperature. The upper oxycline (UO), which begins just below the thermocline, has decreasing oxygen with depth and may be abrupt or broad. The OMZ core has the lowest oxygen values stable over a depth range; at these locations OMZ core oxygen was often < ~1-2 μM, but was always detectable. The lower oxycline (LO) starts at the inflection point at the base of the OMZ core where oxygen begins to increase with depth. The LO is often characterized by a sharp subsurface peak in zooplankton biomass and abundance and the presence of specific

indicator taxa. Oxygen would continue to increase below our sampling range.

The lowest oxygen values in the OMZ core during these cruises occurred at CRD and TB (values down to 1.0 μM). CRD and TB had very thin mixed layers, with the start of the thermocline at 16-34 m in different tows and with temperature and oxygen decreasing rapidly below. However, TB in 2007 had a subsurface oxygen intrusion in the upper oxycline zone, so the OMZ core

did not begin until about 350 m and extended to 550 m, similar to the OMZ core extent at CRD that same year. TB in 2008 had the most vertically extensive OMZ core (oxygen values of 1.1-1.5 μM) extending from depths of 80-700 m, with an abrupt thermocline and almost nonexistent upper oxycline. It was also notable for the occurrence of double subsurface fluorescence peaks, one near the shallow thermocline (~40 m) as expected and also a deeper peak at ~120 m within the low oxygen water of the OMZ core.


The mixed layer extended deeper at the OC and SKQ stations. Fluorescence peaks near the thermocline were deeper (~80-100 m) at OC and SKQ, compared to the thermocline-associated peak (~50 m) at CRD. At OC and SKQ, the oxygen profile was more rounded in shape without a clear cut extensive OMZ core. The upper oxycline was broad, with relatively high oxygen concentration down to ~450 m. The OMZ core was narrow (only ~25 m thick, ~525-550 m at OC and at more variable depths

between 400-600 m in different tows at SKQ). Oxygen values in the core at OC reached only as low as 3.6 μM, substantially



higher oxygen than at CRD or TB. SKQ had some low oxygen values of 1.0 µM for parts of particular net intervals, especially in the mesopelagic horizontal tows during transects through submesoscale features (Wishner et al., 2018).

Temperature decreased with depth below the thermocline and was similar through the mesopelagic at all these stations (Fig. 2). Mixed layer salinity was comparatively low except at the OC and SKQ stations, which were farther north than those of the earlier cruises, and where there was a subsurface salinity minimum (~ 80-250 m), likely an influence of the southern extent of the California Current.

Temperature-salinity and temperature-oxygen graphs illustrate the different water masses and metabolic habitats encountered (Fig. 2, bottom row). The high-resolution temperature-oxygen graph (bottom right) highlights the small scale habitat variability of the OMZ and oxyclines (see also (Wishner et al., 2018). In different locations, animals at the same mesopelagic depth (i.e. same temperature) could experience quite different oxygen regimes; animals tracking a specific oxygen concentration might live at different depths (and temperatures).

### 3.2 Copepod distributions and responses to oxygen profile variability

Copepod distributions are presented in a series of graphs and tables described below. Environmental data (ranges for depth, oxygen, and temperature) and abundances (number $(1000 \text{ m})^{-3}$) in the single net at the depth of maximum abundance (DMA) for each species are in Table 3, along with water column abundances (number $\text{m}^{-2}$ usually for the 0-1000 m depth interval) for that species at that station. Abundances of each species in all tows and nets are in Table S1 (Wishner and Outram, subm.).

### 3.2.1 OMZ oxycline and OMZ core species that shift depth and track oxygen

These were primarily mesopelagic species, most with no diel vertical migration, whose peak abundance was associated with the inflection point at the upper edge (UO) or lower edge (LO) of the OMZ core. These species showed distributional shifts associated with the changing depth of the inflection points and the OMZ core on the different cruises. Peak abundances often occurred abruptly in a very narrow depth interval (a single or only a few sampling nets) and could be several orders of magnitude higher than in adjacent nets.


*Lucicutia hulsemannae.* This copepod (Markhaseva and Ferrari, 2005), formerly called *L. grandis*, is considered to be an indicator species of the LO (Wishner et al., 2000, 2013). Its peak abundance occurred at the LO inflection point at the base of the OMZ core in the tows with lowest oxygen and most extensive OMZ (Fig. 3, Table 3). Its maximum abundance in a single net (997 individuals $(1000 \text{ m})^{-3}$) occurred at 600-625 m at CRD in 2007, with oxygen in that interval at 1.7-5.9 µM and

temperature at 6.3-6.5°C. It was also very abundant at TB in 2008 (maximum of 781 individuals $(1000 \text{ m})^{-3}$), but in this case that peak occurred deeper (800-825 m) and thus at lower temperature (5.5-5.6°C), just below the extensive OMZ core of lowest oxygen. This net sampled a more constrained oxygen range (1.8-2.4 µM) confirming that this species can thrive in extremely low oxygen water (Wishner et al., 2018). At OC and SKQ, where oxygen was not as low and where the oxygen profile was more rounded without a clear-cut OMZ core, *L hulsemannae* was distributed more broadly throughout the overall OMZ and both

oxyclines, and its water column abundance (7-16 $\text{m}^{-2}$ from 0-1000 m) was usually lower. Maximum abundances (65-92 individuals $(1000 \text{ m})^{-3}$) were also substantially less (with one exception) and occurred shallower (450-550 m at OC; 500–600 or 600-650 m at SKQ) at these northern stations, although temperature at depth was similar to CRD. During SKQ in horizontally-sequenced tows reported elsewhere, this species also had strong physiological and distributional responses to very small (several





µM) oxygen differences and had higher abundances in lower oxygen (Wishner et al., 2018). In the SKQ "mid-depth comparison" series, highest abundances were usually associated with the tow and nets with lower oxygen (Fig. 4). Younger stages, from copepodite 2 to 5, were recorded, and often abundant, in many of the same nets and depths as the adults. No adults or young stages were found in shallow samples. *L. hulsemannae* thus appears to be actively growing and developing within this extremely hypoxic part of the OMZ.

A congeneric species, *Lucicutia ovalis*, occurred in the deeper LO several hundred meters below *L. hulsemannae* and consequently at lower temperature and higher oxygen levels (Table 3, not graphed). It also shifted depth between locations and was deeper at CRD (1100-1200 m) and TB (900-1000 m) (stations with a thick OMZ) than at OC (700-750 m). It was less abundant than *L. hulsemannae* and was not found on the SKQ cruise (perhaps due in part to its small size and likely absence from the large size fraction that was sorted).


*Disseta palumbii.* This widely distributed species in the family Heterorhabdidae (Razouls et al., 2005-2019) was also typical of the LO, but its abundance peaked about 50-100 m deeper than *L. hulsemannae* (Fig. 3). Consequently, the oxygen at its DMA was slightly higher and the temperature lower than for *L. hulsemannae*. Its water column abundance and maximum abundance were highest at the stations where OMZ oxygen was higher, and it was more abundant at OC and SKQ than in earlier cruises.

Younger stages (copepodites 4 and 5) were present in some cruises at similar depths to adults and were sometimes more abundant than adults.

Another Heterorhabdidae species complex, *Heterostylites longicornis/longioperculis*, was very abundant from the UO through the OM to the LO and was another characteristic species of this habitat (Fig. 3). It was likely a mixture of two species that are

difficult to separate microscopically (Razouls et al., 2005-2019), so was treated as one entity. In contrast to *D. palumbii*, abundances were highest in the earlier cruises at CRD and TB08, where it co-occurred with *L. hulsemannae* at the LO in very low oxygen. At OC and SKQ where it was less abundant, it occurred primarily in the UO and shallower than *L. hulsemannae* and *D. palumbii*. Oxygen values in its habitat at SKQ were similar to the low values of earlier cruises, but in OC its shallower peaks (100-250 m) were at higher oxygen. Younger stages occurred at many of the same depths.


Two Aetideidae species, *Gaetanus kruppi* and *G. pseudolatifrons*, and two Augaptilidae species, *Euaugaptilus magnus* and *E. nodifrons*, also demonstrated distributional separation between congeners within the LO zone (Fig. 5). *G. kruppi* occurred shallower than *G. pseudolatifrons* and usually at lower oxygen and higher temperature because it was closer to the LO inflection point. Both were most abundant at CRD. *E. magnus* was clearly a LO species, while *E. nodifrons* had a distribution pattern split

between the UO and LO. These latter two species were identified only from the later cruises.

Several other species, including *Metridia brevicauda, Metridia princeps,* and *Paraeuchaeta californica*, also showed a split distribution pattern with peaks of adults usually at both the UO and LO sides of the OMZ core but almost no specimens in the OMZ core itself or in shallow water (Fig. 6, 7). The predominant peak for *M. brevicauda* adults was typically in the UO, while

the predominant peak for *P. californica, M. princeps,* and *E. nodifrons* could be either in the LO or UO. *P. californica* and *E. nodifrons* in the LO occurred deeper and at higher oxygen than the LO secondary peak of *M. brevicauda*. The LO peak (at 675-700 m) for *M. princeps* was especially clear-cut at CRD since this species was not present shallower there; however, its highest total water column abundance was at SKQ where it occurred over a broader depth range at both the UO and LO.



Oxygen where *M. brevicauda* was most abundant was 1.0-5.7 µM with a temperature of 10.0 – 10.8°C (Table 3) in the UO of
CRD in 2007 (300-350 m, maximum adult abundance in this net of 15,958 (1000 m)$^{-3}$), indicating the ability of this species to
thrive at very low oxygen (similar to *L. hulsemannae*) but at somewhat higher temperature. Younger smaller life history stages
of *M. brevicauda* were quantified in the CRD and TB08 cruises (counted only from 0-200 and 550-1000 m in TB08) (Fig. 6), but
not in later cruises, when only the large copepod size fraction (> 2 mm) was sorted (and small taxa like *M. brevicauda* may have
been under-represented). Younger stages (copepodites 2–5) occurred at similar depths as adults, while the youngest stage
(copepodite 1) was very abundant (up to 23,589 (1000 m)$^{-3}$ at night) in TB08, primarily in the lower mixed layer (50-100 m).
This life history distribution pattern suggested active reproduction at depth, with ontogenetic migration of the youngest stages up
into the well-oxygenated warmer mixed layer at some times.

The Scolecitrichidae copepods *Lophothrix frontalis* and *Scaphocalanus magnus* were common UO components during the OC
and SKQ cruises (not differentiated during earlier cruises), with possible DVM within the UO (Fig. 7). Peak daytime abundance
of *L. frontalis* was located at 425-468 m (oxygen 4.6-7.4 µM), while at night it was at 100-150 or 200-250 m where oxygen was
much higher. *Scaphocalanus magnus* occurred somewhat deeper (extending from the UO into the OMZ core) and its possible
DVM was deeper. Both species also had higher abundances in the higher oxygen profile of the SKQ "mid-depth comparison"
series (Fig. 4).

### 3.2.2 *Eucalanus inermis*, a copepod that diapauses in the OMZ

The Eucalanidae copepod *Eucalanus inermis* was especially abundant at CRD and TB where it formed concentrated diapausing
layers at the UO and LO inflection points (Fig. 8). At the LO (when the diapausing layer was fortuitously sampled as an entity in
a single net), it occurred as a monospecific aggregation located just above the more diverse LO community (e.g. *Lucicutia*
*hulsemannae* and associates), as noted earlier (Wishner et al., 2013). Peak abundance of diapausing adults (33,912 (1000 m)$^{-3}$)
was located in a layer at 300-350 m (UO) during CRD in 2007, while at CRD in 2008, the peak abundance (19,242 (1000 m)$^{-3}$)
was at the LO at 500-550 m. At TB in 2007, peak adult abundance occurred at the UO (20-80 m), whereas the next year at TB,
the maximum adult abundance was at the LO at 775-800 m. Multiple layers (near surface, UO, LO) with different life history
stage composition and relative abundances occurred at both locations during both of the first two cruises (Fig. 8); locations of the
diapausing animals were the UO and LO layers while young stages, as well as non-diapausing adults, occurred near-surface.
During the last two cruises, abundances at those more northerly stations were substantially lower (but still included immature life
history stages) and were not as sharply layered, occurring within the broad UO (400-550 m). There was no concentrated layer of
diapausing animals on these later cruises. No DVM was evident for any cruise.

Adult diapausing copepods were primarily females, but large concentrations of copepodite 4 and 5 males and a few adult males,
as well as some younger stages, occurred in these same samples (Fig. 8). During 2008 at TB, there was also a very high
concentration of copepodite 1 and 2 stages (total of 211,894 (1000 m)$^{-3}$) at 60-80 m at the UO, indicating that reproduction and
development had occurred recently. Similar near-surface layers of mixed life stages, including many immature specimens, were
also found at CRD in both years.


Stored oil, an indicator of diapause, was visibly apparent as an orange colored material inside many preserved copepods during
the first two cruises. Individuals from the diapausing layer that had no obvious stored oil were either brownish or clear in





preserved samples. From 67% to 93% of adult (female) *E. inermis* in the diapausing layers (DMA peaks) during these cruises contained stored oil, as did high proportions of the older immature males (copepodite 4 and 5) also present there. Oil presence
was not quantified on the later cruises, but actively swimming specimens from near surface samples examined live at sea were transparent, with some, but not all, having some transparent stored oil.

*E inermis* diapausing layers were located at extremely low oxygen, 1.0-5.7 µM (except for the shallow layer at TB in 2008 where the net probably sampled across zones) (Table 3). These layers were at the edges of the OMZ core, but usually not at the lowest
oxygen of the OMZ. The large percentage of animals with stored oil in these layers, and the sharpness of layer boundaries, were strong indicators that this species undergoes its diapause in a precisely defined zone of extremely low oxygen. Our sampling did not address seasonal cycles at particular locations, so we do not know the temporal progression of reproduction and development or how long diapause layers persisted at depth in low oxygen. The brownish or clear specimens may have been animals that had depleted their oil reserves.

Other common copepod species from the family Eucalanidae observed during the cruises did not have the obvious diapause behavior of *Eucalanus inermis*. *Eucalanus californicus* and *E. spinifer* were moderately abundant farther north during the latter two cruises (not found at CRD and not separately identified at TB) (Fig. 5); they are common in the California Current and subtropics respectively (Goetze, 2003; Goetze and Ohman, 2010). They occurred primarily within the broad UO, with *E.*
*californicus* being slightly deeper and also having many copepodite stages 4 and 5. These species occurred at somewhat higher oxygen than the *E. inermis* diapausing animals.

### 3.2.3 Diel vertical migrators and OMZ intensity

Copepod species in the genus *Pleuromamma* (Family Metridinidae) are well known as strong diel vertical migrators, including species that descend to mesopelagic depths during the day in the ETNP (Haury, 1988; Hirai et al., 2015; Razouls et al., 2005-
2019). Three of these species provided insight into varying strategies in response to differences in OMZ and mixed layer vertical extent and differences in oxygen values among cruises (Fig. 9). We expected that their daytime depth would be most affected by OMZ variability as they coped with different conditions during their diel transit. However, for *P. abdominalis* and *P. quadrungulata*, it was the nighttime depth that changed with the shape of the oxygen profile, while the daytime depth remained similar across cruises for each species (Table 3). For example, for *P. abdominalis*, its nighttime depth was 20-30 m at CRD in
2007 in the high oxygen mixed layer above the sharp thermocline and low oxygen below, but 100-150 m at OC in 2016 and 50-75 m at SKQ in 2017 where the mixed layer was broader and higher oxygen extended deeper. Daytime depth was 250-350 m at all these stations in the UO just above the OMZ core.

*P. johnsoni* (Ferrari and Saltzman, 1998) tended to occur deeper during the day than the other species (400-450 m at CRD in
2007 and SKQ in 2017) and had some day and night layers associated with the LO below the OMZ core (Fig. 9). At TB in 2008, with its vertically extensive OMZ, abundant daytime layers of *P. johnsoni* occurred at both 250-300 m in the OMZ core (oxygen 1.2-1.4 µM, temperature 10.8-11.3°C) and 850-875 m at the LO inflection point (oxygen 2.5-4.3 µM, temperature 5.2-5.4°C). There was also an especially abundant nighttime layer of *P. johnsoni* (20,059 individuals (1000 m)[-3]) at the thermocline where oxygen and temperature were higher (50-60 m, oxygen 21.8-82.6 µM, temperature 19.7-24.2°C).





There were also clear-cut geographic differences in presence/absence and abundance among these species (Fig. 9, Table 3), with *P. abdominalis* most abundant at CRD in 2007 and *P. quadrungulata* most abundant at OC in 2016 and SKQ in 2017, but absent at CRD in 2007. *P. johnsoni* was most abundant at TB but absent at OC in 2016, whereas the other two *Pleuromamma* species were absent at TB in both years (except for one juvenile specimen of *P. abdominalis*). *Pleuromamma* species were specifically

targeted for identification at all stations and nets, so these presence/absence comparisons are strongly supported. Whether these patterns resulted from distinct environmental preferences associated with habitat availability (oxygen and temperature combinations) or from other temporal and spatial changes or ecological factors cannot be resolved with our sampling.

### 3.2.4 Epipelagic habitat compression and mixed layer species responses

Four abundant species that occurred primarily in the warm well-oxygenated mixed layer or near the thermocline were used to

examine the aerobic habitat compression hypothesis by comparing copepod abundances from cruises with substantial difference in the shape of oxygen profiles. The first two cruises had a narrower aerobic mixed layer habitat with an abrupt transition to the OMZ core, while the last two cruises had a broader aerobic mixed layer and more gradual UO. There were several complications associated with choosing representative epipelagic taxa. Many upper water column tropical copepods were small or hard to identify (i.e. *Clausocalanus spp.*) or occurred primarily as immature stages; these were not a focus of this OMZ

project, and many were not identified to species. During some cruises, the mixed layer was sampled only as a single net interval and not subdivided into multiple strata, so small scale distributional differences there were not defined. Also, these four species, generally smaller in size than the deeper living copepods discussed earlier, may have been relatively under-sampled during the later cruises when a slightly larger mesh size was used and only the >2 mm size fraction sorted; however, this should not affect within-cruise distributional patterns.


The Eucalanidae copepod *Subeucalanus subtenuis*, the Paracalanidae copepod *Mecynocera clausi*, and the Lucicutiidae copepod *Lucicutia flavicornis* were most abundant shallower in the thermocline at the first two stations (20-30 m at CRD in 2007, 40-60 m at TB in 2008), but occurred deeper and in lower abundance within the broader mixed layer (usually 75-100 m) on the later cruises (Fig. 10). Maximum abundances reached 33,647 (1000 m)$^{-3}$ (*S. subtenuis* during CRD in 2007, 20-30 m). There was no

clear signal of DVM for these species although our sampling cannot rule out small scale shifts within the mixed layer. For these abundant epipelagic copepods, the vertical range contraction observed in distributions in the two earlier cruises compared to the later ones was likely a response to habitat compression of the aerobic mixed layer.

Another abundant copepod, the Heterorhabdidae *Haloptilus longicornis* occurred at the base of the mixed layer in the first two

cruises (100-150 m at CRD in 2007 and 60-80 m at TB in 2008 during the day) but substantially deeper within the UO (200-300 m during the day) in the latter two cruises (Fig. 9). Because of the different shapes of the oxygen profiles, oxygen at these very different daytime depths was similar and moderate (13.1-61.2 μM), except at TB in 2008 which had lower oxygen. However, temperature was about 3°C lower at the deeper daytime depths in the latter cruises compared to the shallower earlier ones (Table 3). Additionally, *H. longicornis* exhibited a short DVM up to 75-100 m at night at two of the stations.

## 4 Discussion

### 4.1 Overview



The unique feature of our work is the effort to document precise habitat conditions pertinent to metabolic requirements (oxygen and temperature) at depths where particular species are most abundant, and how the shapes of their vertical abundance profiles are modified in concert with the changing vertical profiles of oxygen in these extreme OMZ habitats.


Although the ETNP copepod fauna and, in some cases, distributional relationships to oxygen have been studied previously in numerous locations and programs (Fernández-Álamo and Färber-Lorda, 2006), the sampling schemes, technology, and scientific foci differed from ours. Chen (1986) analyzed vertical distributions for both individual copepod taxa and large multispecies samples, collected with opening-closing bongo nets, at stations along a transect from Baja California to the equator during the
Krill Expedition. Copepod vertical distributions in the Costa Rica Dome, long known as a productive feature important to fisheries (Fiedler, 2002; Landry et al., 2016), have been reported (Décima et al., 2016; Fiedler, 2002; Jackson and Smith, 2016; Landry et al., 2016; Sameoto, 1986; Vinogradov et al., 1991; Wishner et al., 2013). Copepod vertical distributions from MOCNESS tows near the Volcano 7 seamount in the ETNP (13°N 102°W) were described by Saltzman and Wishner (1997). Many studies also analyzed vertical distributions of some of these species in adjacent subtropical regions, including the central
north Pacific (Ambler and Miller, 1987; Landry et al., 2001; McGowan and Walker, 1979) and the southern extent of the California Current off Baja California (Jiménez-Pérez and Lavaniegos, 2004; Longhurst, 1967) where there is a strong OMZ. Mesopelagic copepod distributions in other strong OMZs worldwide have also been detailed in the Eastern Tropical South Pacific off Peru and Chile (Escribano et al., 2009; Judkins, 1980; Tutasi and Escribano, 2019), the southeastern Atlantic (Bode et al., 2018; Teuber et al., 2013a, 2013b, 2019), and the Arabian Sea (Smith and Madhupratap, 2005; Wishner et al., 2008).


Many earlier zooplankton studies did not collect the comprehensive hydrographic data available from modern electronic CTD and oxygen sensors, did not sample and target specific oxyclines and fine scale vertical strata with precision control, did not extend to deeper mesopelagic depths, or occurred before the recent perspective of OMZ deoxygenation impacts that guides present research (Breitburg et al., 2018). It is also essential to have species-level and life stage identification (not just biomass,
size class, or large taxonomic grouping) to understand the multifaceted interactions and evolutionary adaptations of diverse zooplankton to the strong persistent (but variable) oxygen gradients of OMZs. OMZ oxygen is far more variable spatially and temporally than is generally appreciated, even at mesopelagic depths (Fig. 2), and copepod distributions responded to both large and subtle differences.

The physical oceanographic complexity of oxygen distributions and sources in the ETNP, including spatial and temporal variability, has also long been an active topic of investigation (e.g. (Deutsch et al., 2011; Fiedler and Talley, 2006; Margolskee et al., 2019; Wishner et al., 2018). However, although we compared vertical distribution differences relative to oxygen gradients among locations and cruises, our sampling was not designed to elucidate responses to the many other spatially and temporally variable environmental forces pertinent to this region, such as climate cycles (El Nino-Southern Oscillation) or the dynamic
variability of the major regional ocean currents or deep advective flow (Kessler, 2006).

### 4.2 OMZ, oxycline, and "hypoxiphilic" taxa

Oxycline-associated and OMZ species showed a remarkable ability to alter their vertical distributions to conform to different oxygen profile shapes at different locations. Many taxa had very narrow oxygen habitat ranges and strong layering, indicative of precise habitat preference and exceptional tolerance for low oxygen. These species tended to alter depth (and temperature) while
maintaining a relatively constant oxygen level (Table 3). Important caveats overall are that (1) we focused primarily on adults



(except for species previously noted) and (2) total population distributions, even for adults, often extended over broader depth and oxygen ranges beyond the DMA.

We propose the term "hypoxiphilic" to describe species whose population distributions, especially maximum abundances, are focused in habitats with very low oxygen.  This term is best used in a comparative sense (without an exact oxygen definition) to allow flexibility for use in other regions and ecological situations and with other taxa, since we know little about the *in situ* low oxygen physiology of many species (Teuber et al., 2013b; Thuesen et al., 1998). *Lucicutia hulsemannae* is the most obvious "hypoxiphilic" copepod (both adults and young stages living in very low oxygen at multiple locations here and in prior work). Earlier studies of its gut contents indicated recent omnivory (Gowing and Wishner, 1992), suggesting active feeding, as well as

growth, in this habitat.  Other possible "hypoxiphilic" copepods in this area at this time included *Disseta palumbii*, *Gaetanus kruppi*, *Euaugaptilus magnus*, *Eucalanus spinifer*, diapausing stages of *Eucalanus inermis*, and *Metridia brevicauda* adults, but less is known about the vertical and geographic distributional variability of all life history stages for these species.

*L. hulsemannae* has unique metabolic adaptations for hypoxia tolerance in OMZs, including a reverse temperature response that

results in a nearly constant ratio of environmental:critical $PO_2$ of ~1 to 1.5.  In fact, (Wishner et al., 2018) showed that when this ratio (equivalent to the Metabolic Index; (Deutsch et al., 2015)) approaches 2, abundance declines.  A ratio below 1 suggests that the species is incapable of meeting its resting needs, whereas a ratio above 1 allows a proportional increase in the scope for aerobic activities in support of locomotion, growth and reproduction (Wishner et al., 2018).  The minimum $PO_2$ at the minimum temperatures reported at all stations results in a drop in the Metabolic Index consistently below 1.  Although our sampling regime

cannot pinpoint the precise conditions under which individual animals are found, the Metabolic Index suggests that this species cannot tolerate the most extreme low oxygen at low temperature.

### 4.3.  Diapausing in the OMZ

*Eucalanus inermis* is endemic to the ETP, occurring both north and south of the equator (Chen, 1986; Fleminger, 1973; Goetze, 2003, 2010; Goetze and Ohman, 2010; Hidalgo et al., 2005b, 2005a; Jackson and Smith, 2016; Sameoto, 1986).  It is known to

inhabit extremely low oxygen water (Boyd et al., 1980; Escribano et al., 2009; Judkins, 1980; Saltzman and Wishner, 1997; Wishner et al., 2013), as well as oxygenated depths, and has unique physiological and biochemical adaptations for the low oxygen environment (Cass et al., 2014, 2014; Cass and Daly, 2015).  Overall population distributions at particular locations in this and prior work usually encompassed a range of oxygen concentrations, although most earlier studies had relatively broad sampling strata that did not isolate specific oxygen ranges or did not have electronic oxygen sensors co-located on the

zooplankton sampling systems.  DVM by this species, usually over a relatively short depth range between the near-surface and UO, was reported by some authors at some times and places, but not others (Escribano et al., 2009; Jackson and Smith, 2016). We previously identified apparent diapause layers at the upper and lower OMZ boundaries at the CRD and TB stations in 2007-2008 (Wishner et al., 2013), and Saltzman and Wishner (1997), Sameoto (1986), and Vinogradov et al. (1991) also noted abundance peaks associated with OMZ edges in the CRD and other ETNP locations.  Diapause layers in our study varied in

depth at different times and stations, allowing the copepods to remain in a constant low oxygen habitat.

Use of extremely low oxygen habitats for diapause is likely a predator avoidance mechanism during an extended period of inactivity.  Metabolic adaptations that enable longterm survival for copepods in this habitat, while not specifically studied, likely include metabolic suppression and fuel (lipid) storage.  Metabolic suppression, typically triggered by epigenetic mechanisms



facilitated by microRNAs, enables a variety of animals to survive temporary resource limitation and involves shutting down
        energetically costly processes such as translation and transcription, ion transport and protein synthesis (Biggar and Storey, 2010;
        Storey and Storey, 2004). The high proportion of animals with stored oil in these remarkably sharp narrow monospecific layers,
        composed primarily of adult females and older copepodite stages, were strong indicators of diapause. No obvious gut contents
        were visually noted in these samples. *E. inermis* collected at all depths at CRD and TB had stable isotope signatures indicative

of feeding at the surface and not on deeper material (Williams et al., 2014), suggesting inactivity at depth. In an earlier study in
        the same general region and depth (but not noted if individuals were diapausing), transmission electron microscope analyses of
        *E. inermis* gut contents included some tissue and copepod gut epithelium, not visible with light microscopy (Gowing and
        Wishner, 1992). The seasonal ontogenetic and reproductive cycle of *E. inermis* was documented at a location off Chile but the
        sampling depth was too shallow (75 m) to elucidate deeper diapause behavior (Hidalgo et al., 2005a, 2005b). Another calanoid

copepod, *Calanus pacificus*, also has an apparent diapause layer just above the anoxic bottom water of the deep Santa Barbara
        Basin off California (Alldredge et al., 1984; Osgood and Checkley Jr, 1997), and *Calanoides carinatus* in the Arabian Sea and
        Benguela Current diapauses at depth just below the extensive OMZ (Auel and Verheye, 2007; Smith et al., 1998; Smith, 1982;
        Wishner et al., 2008).

### 4.4 Diel vertical migration into low oxygen

Species in the genus *Pleuromamma* have been studied intensively in the ETNP with regard to their extensive DVM into low
        oxygen water, as previously noted. However, unlike resident oxycline and OMZ core taxa, these species appeared to require a
        period of time at night in well-oxygenated water, presumably to pay off the oxygen debt incurred during daytime residence in
        low oxygen. It was surprising that the part of their diel distribution that changed with differing oxygen profiles was the shallow
        nighttime depth, not the deep daytime depth. They continued to migrate to the same depth in the day at different locations,

presumably determined by light penetration, and likely an avoidance response to visual predators (Wishner et al., 2013). At
        night, it seemed as if they came up only until they encountered the depth at which there was sufficient oxygen (but with lower
        temperatures than at shallower depths), within or just above the thermocline. The thermocline was also usually a peak of
        zooplankton biomass (Wishner et al., 2013) and fluorescence (Fig. 2), so food was abundant there as well. Thus, the upper
        nighttime depth likely resulted from the combination of higher oxygen, food availability, and avoidance of higher temperature in

shallower waters where metabolic costs would be elevated.

        *Pleuromamma* species traveled over different depth ranges within low oxygen water to their daytime depth at the different
        stations. Our sampling could not resolve whether they moved faster or took longer to make the journey when the depth range
        was greater, i.e. how long an individual resided in the low oxygen water in different circumstances and what the comparative

metabolic costs might be for the migration. It is also unknown whether all of these species undergo metabolic suppression when
        in low oxygen, as do many euphausiids and other strong migrators (Seibel et al., 2016, 2018). Teuber et al. (2013b, 2013a, 2019)
        found that *Pleuromamma* species in the eastern tropical Atlantic had high thermal and hypoxia tolerances and classified these
        species as adaptive migrants on the basis of functional traits. Active transport by diel vertical migrators is a critical mechanism
        of export flux in the CRD (Stukel et al., 2018), and export fluxes appear to be relatively inefficient in locations with OMZs

(Berelson et al., 2015). Strong presence/absence differences among stations in our study suggested that some locations may have
        exceeded the metabolic capabilities of particular species of diel vertical migrators, which could be a factor affecting export flux.

### 4.5 Aerobic habitat compression and epipelagic copepods





The aerobic habitat compression hypothesis was supported by the distributional shifts of several abundant epipelagic species that occurred over a broader depth range at stations where the well-oxygenated mixed layer extended deeper. However, our sampling
strata, which focused on the mesopelagic, did not resolve small scale details, such as short DVM within the mixed layer or thermocline. Additionally, our taxonomic focus on adults of larger easily identified species excluded numerous smaller epipelagic copepods and immature stages. Higher resolution mixed layer sampling and more taxonomic breadth is required to thoroughly address this issue for ETNP copepods. The high oxygen concentration within the mixed layer would not be a metabolic constraint, but, of course, many other ecological factors could affect distributions.

**4.6 Metabolic implications**

Animal metabolic rates are highly variable depending on physical factors, including temperature and oxygen, as well as the ecological and evolutionary requirements for energy to support growth, reproduction, predator-prey interactions and basic maintenance. The factorial aerobic metabolic scope (FAS, the maximum as a factor of minimum metabolic rate) typically ranges from about 6 at the cold end of a species habitat range, to ~2 at the warmer end (Killen et al., 2016; Seibel and Deutsch, 2019).
The FAS can be estimated as the ratio of the critical PO$_2$ (P$_{crit}$) for maximum over that for resting metabolism. Seibel and Deutsch (2019) recently showed that the physiological capacity to supply oxygen evolves to support the maximum metabolic rate at the maximum environmental PO$_2$ in a species' environment. This means that the maximum available environmental PO$_2$ is a usually a factor of 2-6 above the critical PO$_2$ for resting metabolism, depending on temperature. For species exposed to air-saturated water, including vertical migrators during their nighttime forays into shallow water, the P$_{crit}$ for maximum metabolic
rate is near air-saturation (21 kPa). Any reduction in PO$_2$ below the P$_{crit}$ for maximum metabolic rate will result in a quantifiable reduction in maximum capacity (1/P$_{crit}$ = 4.7% kPa$^{-1}$). In this light, our observations that the nighttime depth distribution of vertical migrators responded more strongly to variations in oxygen content than did the deeper daytime depth is not surprising. Migrators descend during the day to depths consistent with predator avoidance in low light. If the oxygen at those depths is below the P$_{crit}$, then metabolic suppression is triggered, which dramatically increases tolerance time to low oxygen (Seibel et al.,
565 2014, 2016, 2018).

In contrast, permanent residents of the lower oxycline and OMZ core have much lower P$_{crit}$s and very limited aerobic scope. *Lucicutia hulsemannae*, for example, has the lowest P$_{crit}$ ever measured for an animal (~0.1 kPa at 8°C and 0.3 kPa at 5°C), comparable only to two other known OMZ specialists, the pelagic red crab, *Pleuroncodes planipes* (Quetin and Childress, 1976)
and the shrimp, *Gennades* spp. (Wishner et al., 2018). The minimum oxygen concentration recorded across the habitat for *L. hulsemannae*, assuming it occurred at the maximum observed temperature (7.7°C), would result in a minimum PO$_2$ ranging from 0.77 to 0.30 at different locations. These values are very near the P$_{crit}$ for routine metabolism, thus providing little if any aerobic scope (FAS ranges from ~1-3) for activity, growth or reproduction. The reverse temperature effect observed for *L. hulsemannae* and *Gennades* spp., in which higher temperature results in a lower P$_{crit}$ despite a higher metabolic rate, is adaptive in the lower
oxycline where temperature and oxygen are inversely correlated.

**5 Conclusions**

Individual copepod species demonstrated different distributional strategies in response to present-day variability of oxygen vertical profiles in the ETNP region. We identified sets of species that (1) changed their vertical distributions and depth of maximum abundance associated with the depth and intensity of the OMZ and its oxycline inflection points, (2) adjusted their diel
vertical migration, especially the nighttime upper depth, (3) shifted their depth of diapause, (4) or expanded / contracted their



depth range within the mixed layer and upper thermocline in association with the thickness of the aerobic epipelagic zone (Table 2). Congeners, closely related species in the same genus, often showed similar adaptation strategies that were centered at different depths. These species examples represented a range of calanoid copepod families and trophic ecologies. We expect that similar distributional adaptations to OMZ variability occur across a broad suite of copepod groups, as well as other

zooplankton and fish taxa.

Combinations of environmental oxygen concentration and temperature, two critical determinants of metabolic rates, defined habitats with the highest abundances for each species at each site. In different locations, animals living at, or transiting through, the same depth (same temperature) could experience quite different oxygen regimes: for example, the daytime migration paths

of *Pleuromamma* species. Animals requiring a specific oxygen partial pressure might be forced to live at different depths (and temperatures): for example, the lower oxycline copepod *Lucicutia hulsemannae* and thermocline inhabitant *Haloptilus longicornis*. Thus, animals at locations with different oxygen profiles experienced different oxygen regimes, but the same temperature, at a particular mesopelagic depth, while animals whose peak distributions varied in depth experienced different temperature regimes often at the same oxygen level. Previous work demonstrated that zooplankton can and do respond to very

small oxygen and temperature differences in both oceanic and coastal locations (Pierson et al., 2017; Roman et al., 2019; Svetlichny et al., 2000; Wishner et al., 2018). Of course, many other ecological and environmental conditions affect habitat choice and population abundance, but it is likely that the extremely low oxygen of the ETNP OMZ is a critical controlling factor in this region.

As is becoming apparent from numerous studies, the upper ocean to mesopelagic depth range is a complex interwoven ecosystem with intricate relationships among its various inhabitants and their environment. It is also a critically important zone for oceanic biogeochemical and export processes that can be affected by oxygen gradients (Stukel et al., 2018) and hosts key food web components for commercial fisheries. Among the zooplankton, there will likely be winners and losers as ocean deoxygenation continues in the future and becomes more widespread. Changes in individual copepod species abundances,

vertical distributions, and life history strategies may create perturbations to these intricate food webs and processes. For example, "hypoxiphilic" taxa, such as *Lucicutia hulsemannae*, as well as the other oxycline and OMZ species, might increase in abundance and expand their depth range until other physiological (temperature) or ecological (food, predators) barriers become overwhelming. Present day variability provides a window into future scenarios, but much more research is required to fully elucidate and quantify consequences.


**Data availability:** All data needed to evaluate the conclusions in the paper are present in the paper and/or the Supplementary Materials Table S1 (Wishner and Outram, subm.). Upon publication, the full manuscript will be deposited in the BCO-DMO database (with a doi identifier), as required for National Science Foundation-funded projects. Extensive files of continuous hydrographic data from MOCNESS tows are available from KFW. Additional data related to this paper may be requested from

the authors.

**Author contributions:** KFW led the writing effort, with substantial contributions from all co-authors. KFW directed the MOCNESS sampling component including analyses of zooplankton abundances and distributions, with assistance at sea and in the lab by all co-authors. DO sorted and identified most of the copepods and processed and graphed data. BAS directed the

metabolic sampling and experiments at sea and led the analyses and writing of those sections of the paper.



**Competing interests:** The authors declare that they have no conflict of interest.

**Acknowledgments**

We thank the captains, crews, and marine technicians of all the ships and the institutional marine offices (University of Miami, Woods Hole Oceanographic Institution, University of Washington, University of Alaska). MOCNESS instruments and technical support were provided by the University of Miami and Scripps Institution of Oceanography, with assistance from E. Horgan, J. Lovin, P. Wiebe, C. Matson, and J. Calderwood. K. Daly and B. Seibel served as cruise chief scientists. Many students, technicians, and colleagues helped at sea and in the lab over the years with the MOCNESS tows, sample processing, and data

analyses, especially T.J. Adams, N. Charriere, K. Daly, C. Flagg, S. Frazar, J. Graff, J. Ivory, A. Maas, J. McGreal, M. McNamara, M. O'Brien, D. Moore, J. Pelser, B. Phillips, S. Riley, C. T. Shaw, and R. Williams. Funding was provided by National Science Foundation grants OCE0526545 (Daly), OCE0526502 (Daly, Wishner, Seibel), OCE1459243 (Seibel, Wishner, Roman), and OCE1458967 (Deutsch). The University of Rhode Island's Graduate School of Oceanography's summer REU program, SURFO, funded S. Frazar and S. Riley (OCE-0851794, OCE1460819). We also thank our institutions for faculty

and student funding.

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



**Figure Captions**

**Figure 1:  Map showing station locations**.

**Figure 2: Hydrographic data**.  **Top row**: Hydrographic profiles for 0-1000 m.  Each station (colors) is represented by one profile for each variable; all stations are included for that variable in one graph.  Stations are noted by abbreviation and year. Two versions of the oxygen profiles are shown:  a full range version to encompass oxygen from the surface to depth (top left) and a higher resolution version with a smaller oxygen range to highlight the OMZ core and oxycline variability between stations (top row, second from left).  **Second row**:  Similar profiles for the upper water column (0-200 m).  Only a single oxygen range is

used for this well-oxygenated part of the water column.  **Bottom row**:  Temperature-salinity diagram (bottom left) and two oxygen-temperature diagrams, the first (bottom middle) encompassing the full range of these variables and the second (bottom right) highlighting the smaller oxygen and temperature ranges and variability associated with the OMZ core and oxyclines at the different stations.

**Figure 3:  Vertical distributions relative to oxygen:  Lower oxycline species**.  Vertical distributions day (red) and night (dark blue) relative to the oxygen profile (light blue) at each station from 0–1000 m.  Stations are noted at the top (station abbreviation and year), and each column comes from the indicated station.  Each row is a different species.  Missing boxes are locations and cruises where the species was not recorded. "No Data" indicates that the species was either not present or not identified to species in those samples.  "Absent" indicates that the species was searched for but not found (abundance = 0).  Abundances

(number (1000 m)$^{-3}$) are for adults (combined females and males) and come from nets and tows listed in Table 1.  Abundances are plotted at the mid-depth of the sampling interval.  Horizontal abundance axes vary.  Table S1 (Wishner and Outram, subm.) provides numerical abundance data for each net.

       **Figure 4:  Vertical distributions of 3 species relative to oxygen** for the 3 sequential tows (2 day, 1 night) in the "**mid-depth
comparison**" series at SKQ 17.  Oxygen profiles from each tow (color coded by tow, see legend) are shown in the left graph. Species distributions from each tow are color coded to match the oxygen profile.  The depth range for these profiles is 350–650 m.  These species responded to the slight variations in oxygen between tows.

       **Figure 5:  Vertical distributions relative to oxygen:  Congeneric species from 3 genera**.  Each row shows 2 species (adults)
from a particular genus at the same stations (listed at the top and repeated for each half of the figure).  Each row is a different genus.  See Fig. 3 caption for more explanation.

       **Figure 6:  Vertical distributions relative to oxygen for *Metridia brevicauda*** (adults) at each station (top row).  Second and third rows:  Life history stage distributions (color coded bar graphs) are shown for CRD in 2007 night and day for the full depth
sampled and for TB in 2008 for the upper water column (0-200 m, the only depth range with life history stage counts for that cruise) night and day.  Abundance axes vary.  See Fig. 3 for more explanation.

       **Figure 7:  Vertical distributions relative to oxygen:  Upper oxycline species and those with split oxycline distributions.** Two species are shown in each row with stations repeated for each half.  See Fig. 3 for more explanation.






**Figure 8: Vertical distributions relative to oxygen for** *Eucalanus inermis* adults day and night at each station (top row), and daytime **life history stage distributions** (color coded bar graphs) with depth at each of these stations (two bottom rows). Abundance axes vary. See Fig. 3 for more explanation.

**Figure 9: Vertical distributions relative to oxygen: Diel vertical migrators.** Each row is a different species. For the *Pleuromamma* species, which were searched for and identified in all tows, missing figures represent locations with zero abundances. See Fig. 3 for more explanation.

**Figure 10: Vertical distributions relative to oxygen: Epipelagic species.** Each row is a different species. The depth axis for 865 these graphs is 0-200 m. The uppermost point in the graphs includes the sampling interval that extends to the sea surface (plotted at the mid-depth of the interval). See Fig. 3 for more explanation.

**Supplementary Material**

**Table S1:** Details for each net in a tow including depth range and volume filtered, followed by abundances in each net (number (1000 m)$^{-3}$) for each of the species considered in this paper (Wishner and Outram, subm.). Species names are shortened; full 870 names are in Table 2.





**Table 1**. Cruises, stations, tows, and net depth intervals used in copepod abundance graphs. Several tows were often combined to obtain detailed vertical profiles to depth (see sect. 2.1). D=day, N=night. NS = no sample. Table S1 provides data for each net.


| Cruise and Stations | Station Abbrev | Time D/N | Tow IDs | Dates GMT | Depth m | Net Intervals m |
|---|---|---|---|---|---|---|
| **18 Oct -17 Nov 2007** *R/V Seward Johnson* (SJ07) | | | | | | |
| Tehuantepec Bowl | TB | D | 607 | 28 Oct | 0-1200 | 0-20-80-150-350-550-750-900-1200 |
| Costa Rica Dome | CRD | N | 615, 617, 619, 621 | 8-11 Nov | 0-1200 | 0-20-30-40-50-60-80-100-150-200-250-300-350-400-450-500-550-550-750-800-850-900-950-1000-1100-1200 |
| Costa Rica Dome | CRD | D | 616, 618, 623 | 8-12 Nov | 0-775 | 0-20-30-40-50-60-80-100-150-200-250-300-350-400-450-500-550-575-600-625-650-675-700-733-775 |
| **8 Dec 2008-6 Jan 2009** *R/V Knorr* (19502, KN08) | | | | | | |
| Tehuantepec Bowl | TB | D | 626, 630, 631 | 15, 17-18 Dec | 0-1000 | 0-20-30-40-50-60-80-100-150-200-250-300-350-400-450-500-550-700-775-800-825-850-875-900-1000 |
| Tehuantepec Bowl | TB | N | 628, 632, 633 | 17, 19-20 Dec | 0-1000 | 0-20-30-40-50-60-80-100-150-200-250-300-350-400-450-500-550-700-775-800-825-850-875-900-1000 |
| Costa Rica Dome | CRD | D | 635, 637, 640 | 28-30 Dec | 0-1000 | 0-20-30-40-50-60-80-100-150-200-250-300-350-400-450-500-525-550-575-600-625-650-750-900-1000 |
| Costa Rica Dome | CRD | N | 636, 638, 641 | 29-30 Dec, 1 Jan | 0-1000 | 0-20-30-40-50-60-80-100-150-NS-200-250-300-350-400-450-500-NS-525-550-575-600-625-650-750-900-1000 |
| **16 Apr-6 May 2016** *R/V Oceanus* (OC1604B) Oceanus | OC | N | 707, 709 | 26, 28 Apr | 0-1000 | 0-100-150-200-250-300-350-400-450-550-600-650-700-750-800-850-1000 |
| | OC | D | 708, 710 | 26-29 Apr | 0-1000 | 0-100-150-200-250-300-350-400-450-550-600-650-700-750-800-850-1000 |
| | OC | D | 711 | 30 Apr | 350-470 | 468-460-463-384-390-390-367-385-348 (Physiology Tow) |
| | OC | D | 713 | 2 May | 600-800 | 600-625-650-675-700-725-750-775-800 |
| **19 Jan -15 Feb 2017** *R/V Sikuliaq* (SKQ201701S) Sikuliaq | SKQ | N | 716 | 26 Jan | 0-1000 | 0-100-200-300-400-500-600-700-800-1000 |
| | SKQ | D | 718 | 28 Jan | 350-650 | 350-400-425-450-475-500-550-600-650 (Mid-Depth Comparison) |
| | SKQ | D | 719 | 29 Jan | 350-650 | 350-400-425-450-475-500-550-600-650 (Mid-Depth Comparison) |
| | SKQ | N | 720 | 31 Jan | 350-650 | 350-400-425-450-475-500-550-600-650 (Mid-Depth Comparison) |
| | SKQ | N | 721 | 1 Feb | 0-350 | 0-50-75-100-125-150-200-250-300-350 |
| | SKQ | D | 722 | 1 Feb | 0-350 | 0-50-75-100-125-150-200-250-300-350 |
| | SKQ | D | 724 | 4 Feb | 425 | Horizontal Tow (8 nets) |
| | SKQ | D | 725 | 5 Feb | 0-1000 | 0-100-200-300-400-500-600-700-800-1000 |
| | SKQ | D | 726 | 7 Feb | 430 | Horizontal Tow (8 nets) |
| | SKQ | D | 727 | 8 Feb | 600-825 | 600-625-650-675-700-725-775-800-825 (Oxycline Tow) |
| | SKQ | D | 728 | 9 Feb | 800 | Horizontal Tow (8 nets) |




**Table 2.** Copepod species discussed in this paper.

**OMZ and oxycline species**
*Disseta palumbii*
*Euaugaptilus magnus*
*Euaugaptilus nodifrons*
*Eucalanus californicus*
*Eucalanus spinifer*
*Gaetanus kruppi*
*Gaetanus pseudolatifrons*
*Heterostylites longicornis/longioperculis*
*Lophothrix frontalis*
*Lucicutia hulsemannae*
*Lucicutia ovalis*
*Metridia brevicauda*
*Metridia princeps*
*Paraeuchaeta californica*
*Scaphocalanus magnus*

**OMZ diapause species**
*Eucalanus inermis*

**Diel vertical migrators into OMZ**
*Pleuromamma abdominalis*
*Pleuromamma johnsoni*
*Pleuromamma quadrungulata*

**Epipelagic and thermocline species**
*Haloptilus longicornis*
*Lucicutia flavicornis*
*Mecynocera clausi*
*Subeucalanus subtenuis*




**Table 3.** **Depth of maximum abundance** (DMA) for the species (adults) discussed in the text along with environmental data from MOCNESS sensors for that habitat (oxygen and temperature ranges for the DMA net), abundances in the DMA net (Max Abd) (adults, number (1000 m)$^{-3}$), and water column abundances (adults, number m$^{-2}$, 0-1000 m depth interval except 0-750 m
for CRD07 day) for that species at that station. Species names are shortened (full names in Table 2). Data are included for each location for both day (D) and night (N) when available. "X" indicates situations where the species was not recorded, either because it was not present or not identified. "None" indicates situations where the species was searched for but not found, i.e. abundance of 0 and consequently no associated environmental data. Ranges for depth (DMA, m), oxygen (Ox, µM), and temperature (T, °C) are shown for the indicated DMA net (net IDs in Table S1, first 3 digits are tow number). Ranges rather
than means are given, because each net covered a depth interval, and it is not known where within that interval the animals occurred. NA for oxygen = not available (oxygen sensor malfunction).





| | Disseta palumbii | Euaugaptil magnus | Euaugaptil nodifrons | Eucalanus californicus | Eucalanus spinifer | Gaetanus kruppi | Gaetanus pseudolatif | Heterostylit longi/long | Lophothrix frontalis | Lucicutia hulseman | Lucicutia ovalis | Metridia brevicauda | Metridia princeps | Pareuchta californica | Scaphocal magnus | Eucalanus inermis | Pleuroma abdominal | Pleuroma johnsoni | Pleuroma quadrungu | Haloptilus longicornis | Lucicutia flavicomis | Mecynocer clausi | Subeucala subtenuis |
|---|---|---|---|---|---|---|---|---|---|---|---|---|---|---|---|---|---|---|---|---|---|---|---|
| CRD07_D_DMA Net | 6231 | × | × | none | none | 6232 | 6237 | × | × | 6236 | × | 6165 | 6233 | × | × | 6165 | 6166 | 6163 | none | none | 6181 | 6187 | 6187 |
| CRD07_N_DMA Net | 6198 | × | × | × | none | 6198 | 6193 | 6173 | × | 6173 | 6191 | 6156 | 6194 | × | × | 6155 | 6217 | 6173 | none | none | 6212 | 6216 | 6218 |
| TB07_D_DMA Net | × | × | × | × | none | × | × | × | × | 6072 | × | 6077 | × | × | × | 6077 | 6072 | 6072 | none | × | 6078 | 6078 | 6078 |
| TB08_D_DMA Net | × | × | × | × | × | × | × | 6315 | × | 6315 | 6311 | 6316 | × | × | × | 6316 | 6296 | none | none | 6263 | 6264 | 6265 | × |
| OC_D_DMA Net | 7135 | 7105 | 7081 | 7107 | 7108 | 7081 | 7085 | 7111 | 7077 | 7108 | 7104 | 7117 | 7111 | 7112 | 7112 | 7112 | 7084 | 7083 | none | 7088 | 7088 | 7088 | 7088 |
| OC_N_DMA Net | 7096 | 7093 | 7091 | 7071 | 7071 | 7095 | 7092 | 7077 | 7077 | 7098 | 7094 | 7071 | 7073 | 7073 | 7071 | 7071 | 7077 | 7077 | none | 7074 | 7078 | 7088 | none |
| SKQ_D DMA Net | 7278 | 7273 | 7221 | 7276 | 7184 | 7272 | 7272 | 7194 | 7186 | 7254 | 7196 | 7196 | 7255 | 7274 | 7278 | 7193 | 7221 | 7196 | 7257 | 7227 | 7227 | 7227 | 7227 |
| SKQ_N_DMA Net | 7162 | 7162 | 7162 | 7276 | 7165 | 7162 | 7161 | 7207 | 7213 | 7201 | 7208 | 7208 | 7166 | 7162 | 7162 | 7203 | 7217 | 7217 | 7211 | 7169 | 7216 | 7227 | 7227 |
| CRD07_D_Max Abd | 23 | × | × | none | none | 42 | 0 | 1004 | × | 997.3 | 0 | 15958 | 49 | × | × | 33912 | 40 | none | none | 2210 | 11216 | 33647 | NA |
| CRD07_N_Max Abd | 25 | × | × | none | none | 25 | 21 | 1151 | × | 64 | 75 | 8542 | 17 | × | × | 16539 | 13386 | none | none | 1515 | 8092 | 21864 | NA |
| TB07_D_Max Abd | × | × | × | × | × | × | × | × | × | 288 | × | × | × | × | 5255.4 | 20.80 | 1234.3 | 302.7 | none | 435.4 | 1989.1 | × | × |
| TB08_D_Max Abd | 40.6 | 2.7 | 1.3 | 3.1 | 6.4 | 11.3 | × | 1040.8 | 63.7 | 780.6 | 63.2 | 1962.2 | 3.8 | 14.5 | 4905.5 | 246 | 0 | 435.4 | 10.8 | 0 | 15215.5 | 24.1 | 118.9 |
| OC_D_Max Abd | 24.8 | 1 | 0.8 | 11.5 | 11.5 | 10.9 | 0.8 | 83.2 | 82.8 | 1.2 | 10.5 | 13.1 | 8.1 | 7.7 | 41.6 | 80.2 | 26.7 | 10.9 | 8.9 | 144.8 | 144.8 | 24.1 | 2.4 |
| OC_N_Max Abd | 53.8 | 3.7 | 5.3 | 4.6 | 4.6 | 2.2 | 0.8 | 46.4 | 91.9 | 1.9 | 26.8 | 81.9 | 17.6 | 17.9 | 69 | 31 | 83.2 | 0 | 202.8 | 120 | 4.8 | none | none |
| SKQ_D_Max Abd | 22.1 | 1.1 | 2.2 | 0 | 3 | 7.1 | 8.9 | 20.3 | 44.4 | 65.2 | 3.1 | 3.1 | 3.6 | 4.9 | 226.8 | 288.2 | 382.1 | 38.5 | 6.8 | 10.6 | 5.3 | × | × |
| SKQ_N_DMA | 700-800 | 700-800 | 700-800 | none | 400-500 | 700-800 | 400-1000 | 200-250 | 200-250 | 350-400 | none | 350-400 | 700-800 | 700-800 | 500-550 | 500-550 | 75-100 | 300-350 | 300-350 | 100-150 | 75-100 | 75-100 | 75-100 |
| SKQ_D_DMA | 700-800 | 725-778 | 300-350 | 652-881 | 475-500 | 778-813 | 778-813 | 475-500 | 500-600 | 500-600 | 425-450 | 425-450 | 704-725 | 611-633 | 611-633 | 300-350 | 425-450 | 75-100 | 300-350 | 207-214 | 207-214 | 75-100 | 75-100 |
| OC_D_DMA | 675-700 | 650-700 | 550-600 | 550-600 | 450-550 | 400-450 | 800-850 | 800-825 | 450-550 | 800-825 | 700-750 | 775-800 | 400-450 | 400-450 | 775-800 | 250-300 | 250-300 | 250-300 | 189-228 | 189-228 | 33336.1 |  |  |
| OC_N_DMA | 600-650 | 750-900 | 850-1000 | 400-450 | 450-450 | 650-700 | 800-850 | 200-250 | 450-550 | 450-550 | 700-750 | 367-385 | 460-468 | 460-463 | 460-463 | 300-350 | 100-150 | 100-150 | 250-300 | 207-214 | 207-214 | 0-100 | 0-100 |
| TB08_D_DMA | × | × | × | × | × | × | × | 800-825 | 460-468 | 800-825 | 900-1000 | 775-800 | 400-950 | 400-725 | 775-800 | 250-300 | none | 60-80 | 250-300 | 60-80 | 50-60 | 50-60 | 50-60 |
| TB07_D_DMA | × | × | × | × | × | × | × | × | × | 750-900 | × | 250-300 | × | × | 300-200 | 20-80 | 20-30 | 1.8-7.2 | none | 1.3-2.0 | 20-30 | 0-20 | 0-20 |
| CRD07_N_DMA_Ox | 7.8-8.4 | × | × | none | none | 7.8-8.4 | 550-747 | 1.1-7.6 | × | 1.1-7.6 | 28.9-35.2 | 1.3-1.6 | 14.6-18.7 | 1.1-7.6 | 1.2-1.6 | 1.2-1.6 | 1.1-7.6 | 1.8-7.2 | 5.2-5.9 | 16.7-22 | 8092 | NA | NA |
| CRD07_D_DMA_Ox | 7.5-9.7 | × | × | none | none | 7.4-8.3 | 575-600 | 1.3-2.0 | × | 1.7-5.9 | none | 1.0-5.7 | 7.1-8.0 | 1.0-5.7 | 1.0-5.7 | 1.0-5.7 | 3.4-8.6 | 1.0-1.1 | 8.2-9.1 | 13.1-21.6 | NA | 13.1-21.6 | NA |
| TB08_D_DMA_Ox | × | × | × | × | × | × | × | 1.84-2.41 | 4.6-5.0 | 1.84-2.41 | 5.7-6.9 | 1.4-1.9 | 4.6-5.0 | 6.5-8.2 | 1.4-1.9 | 1.4-1.9 | 1.2-1.4 | 1.2-1.4 | none | 1.3-2.0 | 39.3-74.5 | 28.0-28.0 | 196-200 |
| TB07_D_DMA_Ox | × | × | × | × | × | 18.4-24.8 | × | 1.757-2.20 | × | 1.757-2.20 | × | × | × | × | × | 3.0-200 | 1.1-7.6 | 1.8-7.2 | 10.8-11.3 | 16.7-22 | NA | NA | NA |
| OC_D_DMA_Ox | 5.8-6.3 | 5.4-6.6 | 4.3-4.8 | none | 4.4-5.3 | 5.6-8.2 | 25.6-69.3 | 4.6-5.0 | 63.7 | 4.4-5.3 | 5.7-6.9 | 4.1-10.2 | 4.6-5.0 | 6.5-8.2 | 4.4-5.0 | 1.4-1.9 | 15.7-32.0 | none | 12.7-17.4 | 203-231 | 39.3-35.7 | 39.3-74.5 | 3.9-35.7 |
| OC_N_DMA_Ox | 4.4-5.8 | 7.4-4.9 | 4.1-15.4 | none | 4.1-15.4 | 8.9-10.7 | 97.3-195 | 97.3-195 | 1.2 | 3.6-4.6 | 6.4-7.4 | 7.1-10.2 | 11.7-17.8 | 4.1-5.4 | 4.1-5.4 | 4.4-5.0 | 15.7-32.0 | none | 97.3-195 | 203-231 | 15.8-21.7 | 15.1-21.7 | 3.9-35.7 |
| SKQ_D_DMA_Ox | 4.0-4.8 | 4.2-7.4 | 3.2-5.9 | 3.2-5.9 | 4.2-6.1 | 6.6-9.1 | 1.0-1.7 | 5.0-7.4 | 1.9 | 3.0-6.8 | 1.9 | 1.2-5.7 | 11.7-17.8 | 4.5-6.1 | 6.8-20.1 | 6.8-20.1 | 97.3-195 | 1.2-5.7 | 11.7-15.1 | 189-228 | 15.8-21.7 | 15.1-21.7 | 203-231 |
| SKQ_N_DMA_T | 3.2-5.3 | 3.2-5.3 | 3.2-5.3 | none | 3.4-7.8 | 3.2-5.3 | 8.7-9.0 | 39.5-65.5 | 81.9 | 1.5-3.3 | none | 1.3-4.9 | 1.3-7.1 | 3.2-5.3 | 6.3-21.5 | 6.3-21.5 | 6.8-20.1 | 6.3-21.5 | 207-214 | 207-214 | 170-210 | 202-205 | 203-215 |
| CRD07_D_DMA_T | 5.4-5.7 | × | × | none | none | 5.7-5.9 | 6.5-6.6 | 1.3-2.0 | 10.5-11.5 | 6.3-6.5 | none | 10.0-10.8 | 5.9-6.0 | × | × | 10.0-10.8 | 8.2-9.1 | none | 12.7-13.4 | 15.6-19.2 | 15.6-19.2 | 15.6-19.2 | 20.3-22.7 |
| CRD07_N_DMA_T | 5.7-6.1 | × | × | none | none | 5.7-6.1 | 5.8-7.4 | 5.8-7.4 | 63.7 | 5.8-7.4 | 3.7-4.2 | 10.8-11.6 | 4.7-4.9 | × | × | 10.1-10.8 | 5.8-7.4 | 5.8-7.4 | 12.7-13.4 | 15.6-17.3 | 26.8-26.8 | 27.5-27.7 | 20.3-22.7 |
| TB08_D_DMA_T | × | × | × | none | none | × | 5.8-7.4 | 5.5-5.6 | × | 5.5-5.6 | 4.7-5.1 | 5.6-5.8 | × | × | 5.6-5.8 | 13.5-27.4 | 5.2-5.9 | × | 13.8-14 | 15.6-17.3 | 39.3-74.5 | 15.8-21.7 | 17.2-20.8 |
| TB07_D_DMA_T | × | × | × | × | none | × | × | × | × | 43.2 | × | × | × | × | 348.6 | none | none | × | none | × | 15.8-21.7 | 254.3 | 490.9 |
| OC_D_DMA_T | 5.7-5.9 | 5.7-6.0 | 7.6-8.1 | 6.3-6.6 | 6.6-7.7 | 7.6-8.1 | 10.4-11.0 | 7.5-7.7 | 39.6 | 6.6-7.7 | 4.7-5.1 | 8.7-9.1 | 7.5-7.7 | 7.6-8.1 | 7.7-7.8 | 5.6-5.8 | 9.6-10.4 | none | 14.4-16.9 | 21.1-23.9 | 21.1-23.9 | × | 17.2-20.8 |
| OC_N_DMA_T | 6.0-6.4 | 5.2-5.4 | 7.9-8.3 | 7.9-8.3 | 6.7-7.7 | 7.6-8.1 | 11.7-15.1 | 11.7-15.1 | 10.6 | 6.8-7.7 | 6.4-7.6 | 7.9-8.3 | 8.8-9.4 | 7.9-8.3 | 7.9-8.3 | 7.7-7.8 | none | 302.7 | none | 8.8-9.6 | 15.8-21.7 | 15.1-21.7 | × |
| SKQ_D_DMA_T | 6.2-6.4 | 9.6-9.9 | 9.6-9.9 | 9.8-6.2 | 7.7-7.9 | 5.1-5.2 | 5.1-5.2 | 7.6-8.1 | 11.7-15.1 | 6.8-7.3 | 8.3-8.6 | 8.3-8.6 | 7.3-8.5 | 5.6-5.7 | 6.2-6.4 | 7.0-7.6 | 9.6-9.9 | 8.3-8.6 | 9.4-10.2 | 9.5-9.9 | 19.3-21.4 | 19.3-21.4 | × |
| SKQ_N_DMA_T | 5.4-5.9 | 5.4-5.9 | 5.4-5.9 | none | 7.6-9.0 | 5.4-5.9 | 4.4-4.5 | 8.7-9.0 | 10.5-11.5 | 6.2-6.7 | 9.0-9.4 | 9.0-9.4 | 5.4-5.9 | 5.4-5.9 | 5.4-5.9 | 7.2-7.8 | 22.6-22.7 | 20.3-22.7 | 9.9-11.2 | 19.3-21.4 | 22.6-22.7 | 20.3-22.7 | 20.3-22.7 |
| CRD07_D #/m2 | 0.5 | × | × | 0 | 0 | 2.1 | 2.5 | 78.5 | × | 28.7 | × | 984.3 | 1.2 | × | × | 2182.5 | 217.5 | 2.5 | 12.7-13.4 | 0 | 302.4 | NA | NA |
| CRD07_N #/m2 | 2.7 | × | × | × | 0 | 1.8 | 2.5 | 230 | × | 9.6 | 7.5 | 755.8 | 1.4 | × | × | 1345.5 | 353.7 | 10.9 | 13.8-14 | 122.5 | 62.1 | 117.7 | 608 |
| TB07_D #/m2 | × | 0.7 | × | × | 1.2 | 0.8 | 0.1 | × | × | 43.2 | × | × | × | × | 246 | 0 | 0 | 0 | × | 0 | 179.3 | 125.1 | 118.9 |
| TB08_D #/m2 | 7 | 0.1 | 0.3 | × | × | 0.2 | 0.1 | 60 | 0.9 | 39.6 | 6.3 | 107.8 | 1.3 | 0.5 | 1.7 | 6.5 | 6.5 | 0 | 435.4 | 0 | 171.1 | 254.3 | 490.9 |
| OC_D #/m2 | 7 | 0.1 | 0.2 | 1 | 1 | 0.2 | 0.1 | 1.4 | 0.7 | 10.6 | 0.2 | 0.6 | 1.2 | 0.9 | 6.5 | 6.5 | 6.5 | 0 | 3.7 | 1.8 | 16.1 | 14.5 | 2.4 |
| OC_N #/m2 | 6 | 0.8 | 0.5 | 0.1 | 0 | 0.1 | 0.1 | 4.8 | 2.8 | 10.6 | 0.2 | 2.3 | 1.2 | 0.9 | 2.2 | 2.2 | 2.2 | 13.2 | 12.9 | 12.8 | 4.8 | none |  |
| SKQ_D #/m2 | 4.5 | 0.8 | 0.5 | 1 | 0.6 | 0.5 | 1.1 | 0.4 | 2.8 | 15.6 | none | 0.2 | 4.1 | 0.9 | 4 | 1.4 | 2.5 | 1.4 | 9 | 1.6 | 12.8 | 4.8 | 0.3 |
| SKQ_N #/m2 | 4.8 | 0.1 | 0.2 | 0 | 0.6 | 0.1 | 0.4 | 2.3 | 1.1 | 7.1 | none | 1.1 | 1 | 0.3 | 2.8 | 29.9 | 7.2 | 0.2 | 5.5 | 0.7 | 0.5 | 0.2 | 0.1 |



**Figure 1**

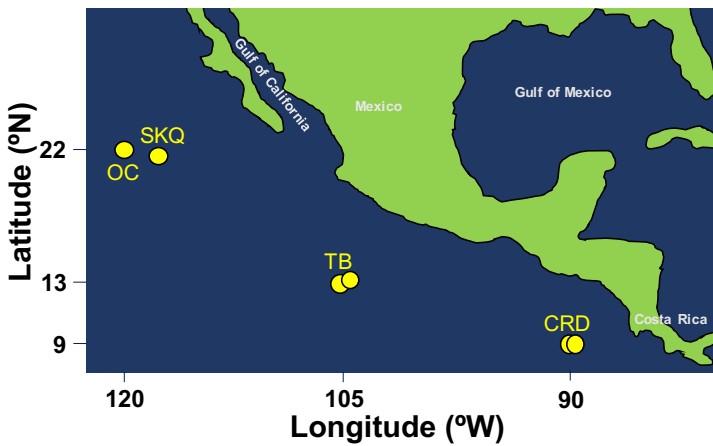



**Figure 2**





**Figure 3**



**Figure 4**

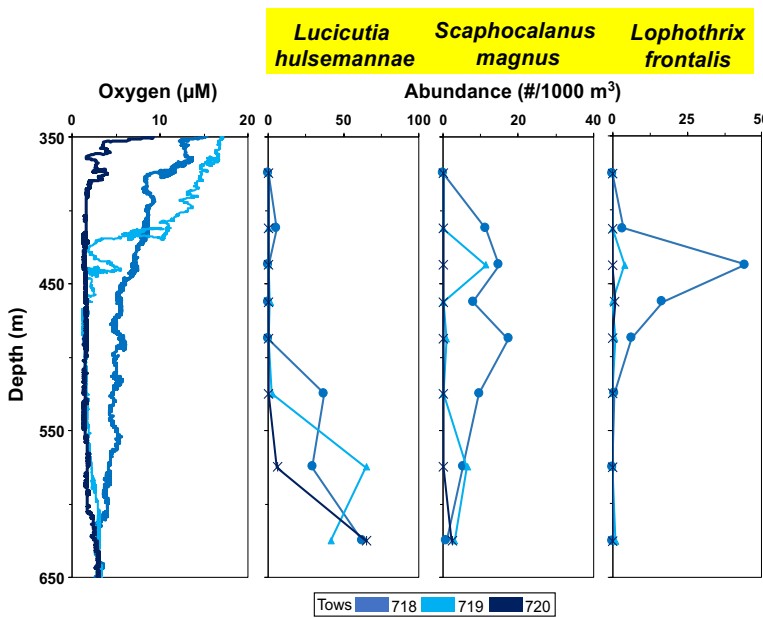



**Figure 5**







**Figure 6**

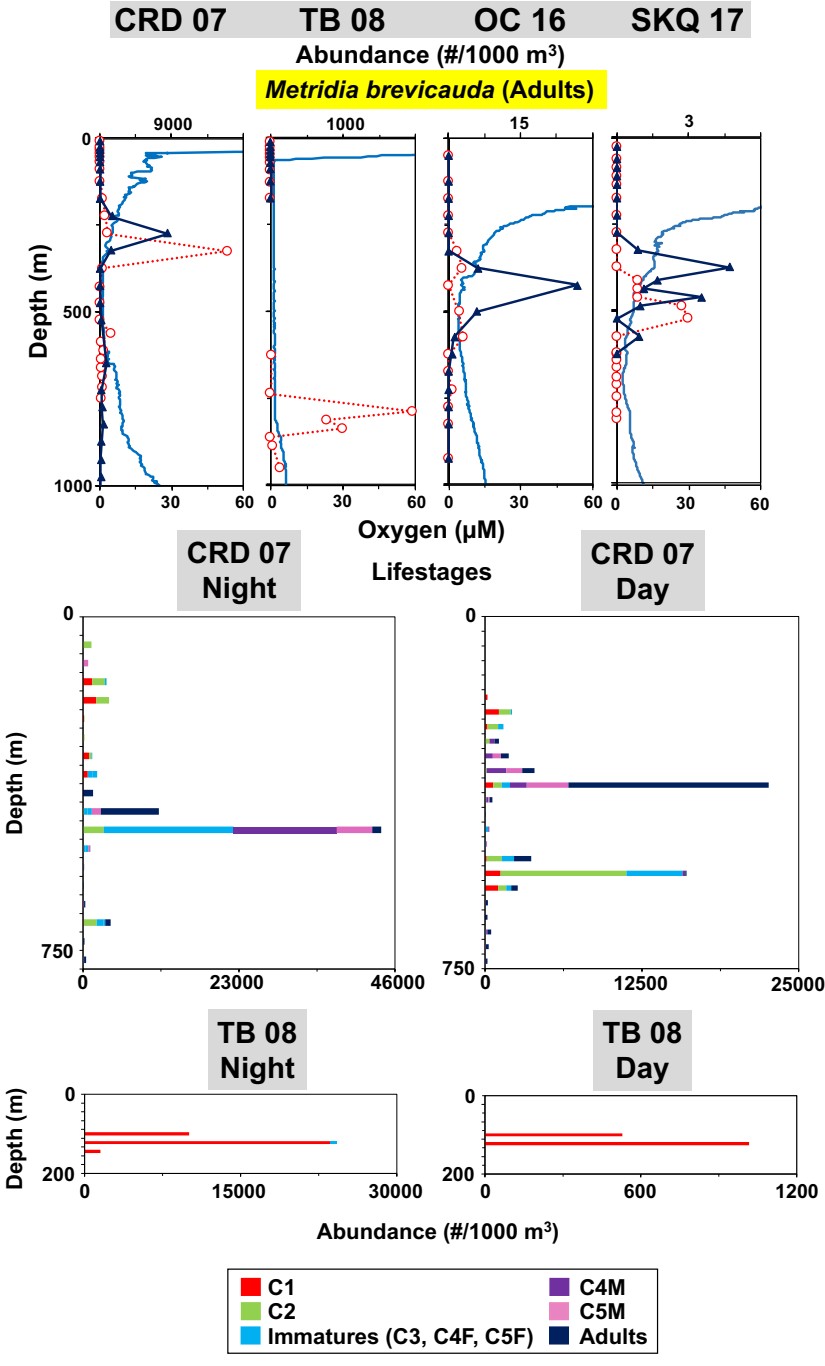



**Figure 7**

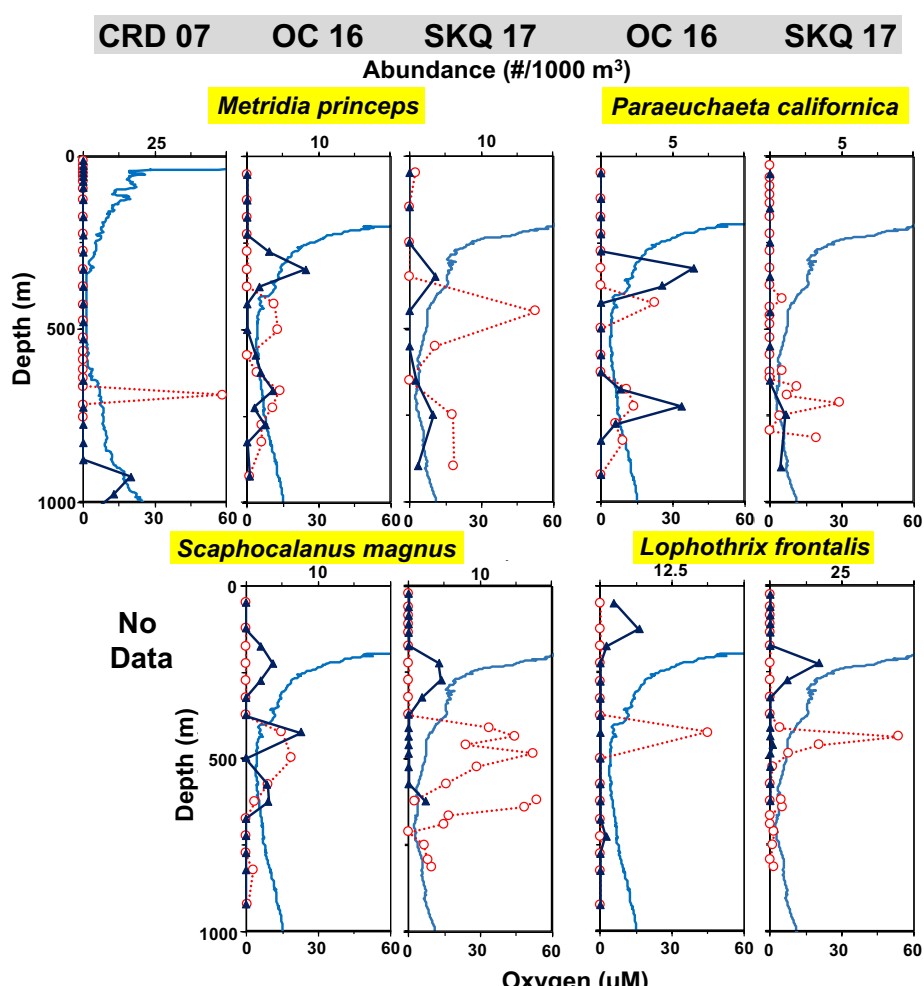



**Figure 8**





**Figure 9**

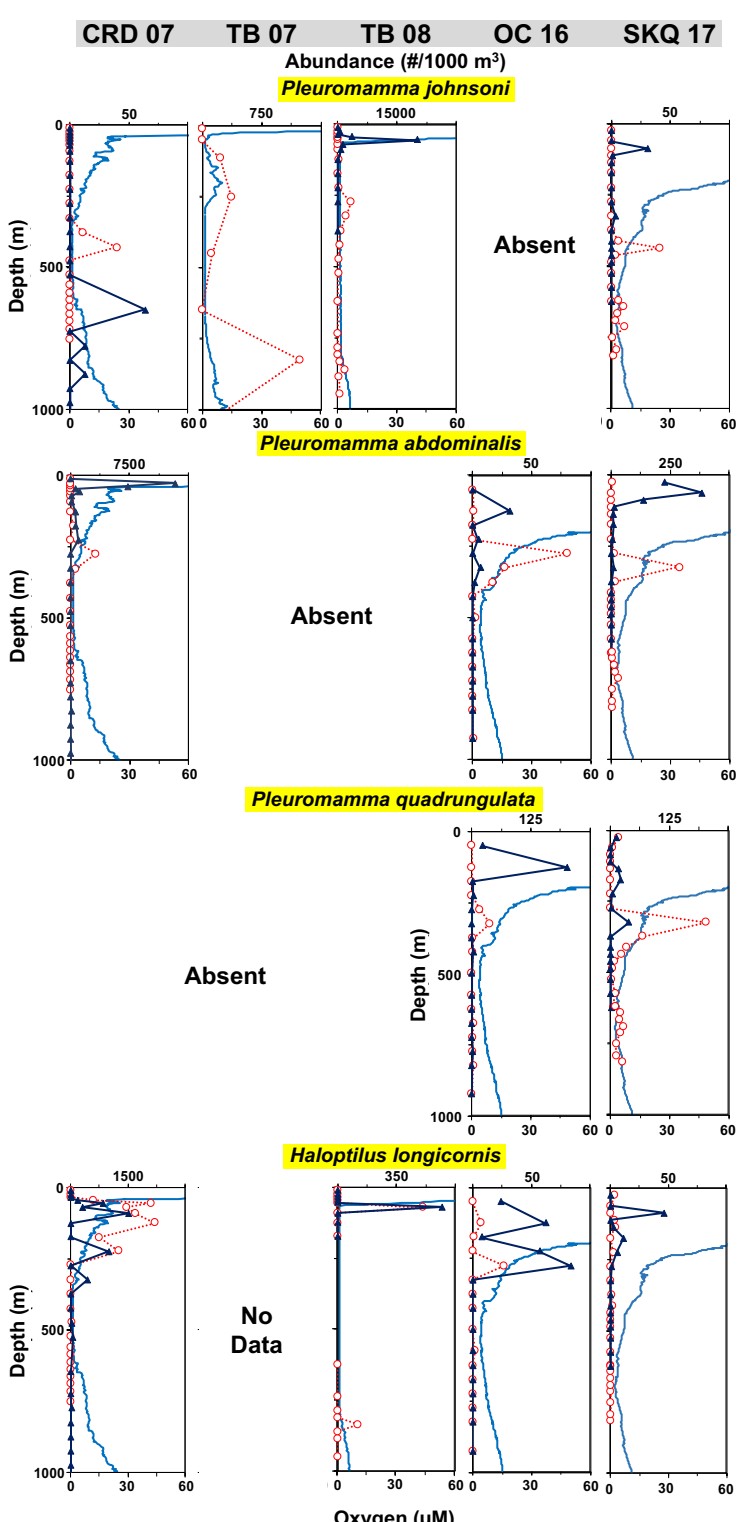





**Figure 10**

