# Peer review of "Ocean Deoxygenation and Copepods: Coping with Oxygen Minimum Zone Variability"

_Biogeosciences, 2019_

## Referee Comment (RC1) · Anonymous Referee #1 · 12 Dec 2019

Ocean deoxygenation and zooplankton: very small oxygen differences matter

This is a large and somewhat complex paper that reports in detail primarily on the depth distribution of a number of different copepod species throughout the ETNP region in areas with prominent and persistent OMZs. The fact that these regions are expanding and that the productivity of the regions is often tied to the extent and magnitude of the deoxygenated area suggests the importance of understanding these regions. Zooplankton as mediators of energy and material flow through planktonic foodwebs are key to this understanding. The authors are well suited to report on the work, having a strong publication record on low oxygen open ocean systems, and the amount of data included in this paper is impressive.

The goals of the paper, to examine the distribution and migration of copepods that

utilize the OMZ, and determine whether those distributions and migrations change with a changing vertical oxygen profile, are useful for understanding habitat use as these habitats change with climate. The text does a good job of answering these questions and describing the data but I think the paper would benefit from a better presentation of the data in tables and figures, and perhaps a summary diagram or two, as well as a summary table. The authors categorize 23 copepod taxa into 4 categories to better understand habitat usage, and while there are observed differences within each category, they are well described by the species included for each category. In addition, there are two basic habitat structures that are compared - one with an abrupt upper oxycline and shallow mixed layer, and another with a more graduate upper oxycline and deeper mixed layer. This would be a useful way to summarize the data: a table and/or diagram showing how each of the 4 categories (generally) responds to both habitats. As noted, this is described well in the text, but it would be especially useful for illustrating the patterns and comparing the graphs that are presented for individual species.

The data that were presented in graphs could be better represented in some cases as well. Figure 2 is somewhat confusing with different vertical and horizontal scales for the same data. The zoomed aspect is a good idea, but I think that the graphs should at least be rearranged, with shallower graphs above deeper one, and perhaps the deeper ones should have a depth minimum at the place where the shallower ones end. Also, a box around the region that is zoomed in could be usfeul, e.g. for the upper layer oxygen data, so that it looks more like an inset. For the TO plots on the bottom, I think a box around the region that is zoomed in will be helpful too. I also wondered about the variability in a given station, because the CTD data is shown as one line from a representative cast, while the zooplankton data is an aggregate. Some comment or presenation of the variability, at least of O2 would be useful. As thin lines of the same color, or a shaded region around the O2 perhaps. This may not be possible and may make the graphs too busy, so even just a note in the text or some supplementary figures would help alleviate concerns about that variability.

For figures 4-10 it might be helpful to have a title to the overall figure, describing which category each group of species belongs to and/or what type of data it represents, simply for quick reference, as they are all very similar. In the published manuscript the caption will be with the figure so this may not be necessary, but because they are all so similar, there might be a way to set them apart.

Some other specific comments:

Table 3 appears unfinished. I think the first column with data names should be reformatted and split into multiple columns for the different metadata and using actual words and not abbreviations (e.g. "D" and "N")

The discussion of oil presence in E. inermis is interesting, but the text presents data not shown in any figures, and it would be better to note that this data is not shown or show it. In particular the inclusion of the percentages of individuals with oil in them should be cited in some way, and perhaps an indication of how many individuals were observed. If this is not a fully quantified number (e.g. if it is anecdotal) it may still be important and worthy of inclusion but it needs some documentation.

---

## Short Comment (SC1) · 13 Dec 2019

We thank Reviewer 1 for perceptive and helpful comments and will work to improve the figures and tables as suggested. More comments will follow, as we digest and respond to individual points in the review.
* * *

---

## Referee Comment (RC2) · Anonymous Referee #2 · 6 Jan 2020

In their manuscript "Ocean Deoxygenation and Copepods: Coping with Oxygen Minimum Zone Variability", Wishner et al. explore the vertical distribution of the copepod community in the Eastern Tropical North Pacific by using D/N paired MOCNESS tows. The strength of the manuscript is also a weakness: it contains quite a lot of high-quality data (which is valuable to the scientific community) but as is, it does not well concatenate information, and a statistical analysis is entirely lacking. Given that mean T, S, $O_2$, Chl-a values are available for each sample, it should be attempted to tease out the main environmental drivers regulating the vertical distribution at day and night, and to present a physiological niche in which the respective species is to be found. Since the metabolic implications are discussed in some detail, I was wondering why environmental oxygen concentrations, rather than $pO_2$, are reported thoughout

the paper. It would be much easier for the reader to understand the contraints, in particular for those species where pcrit data are available (consider extrapolation as a function of temperature). Fig. 1: This map does not reveal much oceanographic information to the reader. Consider including e.g. oxygen contours or average annual surface productivity. Lat/Lon grid should rather be equally spaced (I understand that the goal was to add the approximate lan/lon values for the sampling stations, but the exact values are given in the metadata table, and linear axes make it easier for the reader to visually grasp area size and distances. Fig. 2: This is a very large and very busy figure, mainly due to the many different colors. First, I recommend using the mean profile instead of a chosen single profile for each station (maybe with shaded error, but this might overcrowd the graphs). Second, choose three colors that are the same or similar for the three regions. Try to make the figure fit into a page (lower panels are wider, legend is out of the figure). Oxygen profiles in these would be helpful. Plot area lines could be removed to make some space, but tick marks added because difficult to read with just one tick mark. Figure 4-10: These are way too many figures, they are difficult to read, and they don't convey as much information as they could. Sometimes the panels are organized in a confusing way (e.g. plots from the same area are not next to each other). I suggest to move the majority of these into a supplement, and only keep more integrative figures in the manuscript (which could be, e.g., scatter plots of multivariate analyses or histograms of abundance distribution against oxygen and/or temperature rather than single station profiles). As for the stacked bar charts, I recommend variable bar width so that the bar covers the entire depth stratum sampled as there are no "gaps" between nets (this way, also the colors are more visible). Bar area then is proportional to integrated abundance in the respective depth layer. Day/Night plots of the same station should be scaled the same, and might be mirrored against each other to save space and facilitate comparison. I have added some additional, specific comments to a marked-up version of the pdf.

Please also note the supplement to this comment:

https://www.biogeosciences-discuss.net/bg-2019-394/bg-2019-394-RC2-supplement.pdf

**Supplement:**

[revised manuscript text omitted]

---

## Author Response (AR1)

**Wishner:  Final response to reviewers.  bg-2019-394**.

We appreciate the time and attention of the reviewers.  Below, we discuss our responses to their suggestions and what we intend to do.

**Reviewer 1**

*I think the paper would benefit from a better presentation of the data in tables and figures, and perhaps a summary diagram or two, as well as a summary table.*
*This would be a useful way to summarize the data: a table and/or diagram showing how each of the 4 categories (generally) responds to both habitats.*

We will prepare a summary diagram and consider a summary table.
We added schematic diagram Fig. 11.

*Fig. 2 Reviewer 1:  Figure 2 is somewhat confusing with different vertical and horizontal scales for the same data. The zoomed aspect is a good idea, but I think that the graphs should at least be rearranged, with shallower graphs above deeper one, and perhaps the deeper ones should have a depth minimum at the place where the shallower ones end. Also, a box around the region that is zoomed in could be usfeul, e.g. for the upper layer oxygen data, so that it looks more like an inset. For the TO plots on the bottom, I think a box around the region that is zoomed in will be helpful too.*

*Fig. 2 Reviewer 2: This is a very large and very busy figure, mainly due to the many different colors. First, I recommend using the mean profile instead of a chosen single profile for each station (maybe with shaded error, but this might overcrowd the graphs). Second, choose three colors that are the same or similar for the three regions. Try to make the figure fit into a page (lower panels are wider, legend is out of the figure). Oxygen profiles in these would be helpful. Plot area lines could be removed to make some space, but tick marks added because difficult to read with just one tick mark.*

We will add boxes to make these diagrams easier to understand.  We prefer to have the full water column profiles at the top for overview, and the zoomed upper water column region below, however.  We think that it is important to show individual station profiles, hence the different colors, one for each station. (What are the "3 regions" of Reviewer 2)?  Mean data would obscure the variability and nuances in distributions that we are trying to show (see also next comment below).  The lower panels in Fig 2 are wider (square) because they plot two hydrographic variables (salinity and temperature or oxygen and temperature), not depth.  (We don't understand the "oxygen profiles in these" comment of Reviewer 2). Given the complexity of Fig. 2, we think that more tick marks would introduce too much clutter, but we will evaluate adding some.
We revised Fig 2 with additional labels and indicator boxes.

*I also wondered about the variability in a given station, because the CTD data is shown as one line from a representative cast, while the zooplankton data is an aggregate. Some comment or presentation of the variability, at least of O2 would be useful. As thin lines of the same color, or a shaded region around the O2 perhaps. This may not be possible and may make the graphs too busy, so even just a note in the text or some supplementary figures would help alleviate concerns about that variability.*

Understanding the variability of oxygen is a valid concern. We will provide some supplementary figures in which all oxygen profiles from a particular station are shown to give a sense of variability. At each station, there were multiple MOCNESS tows to different depths that we used to construct the full abundance profiles (see Table 1 and methods text). However, adding all of the oxygen profiles to the abundance profile figures would really clutter them.

We added Supplementary Fig. S1 to show all the oxygen profiles at each station.

The point of the abundance figures is to highlight the depth of maximum abundance (DMA), i.e. where the species is most abundant, and the nuances of their distributions relative to the different shapes of the oxygen profiles. Depth, oxygen, and temperature ranges for the single net at the DMA are presented in Table 3. We believe that ranges, rather than means, are the most valid way to interpret their habitat because MOCNESS tow nets sample over a depth interval and consequently over a range of oxygen and temperature. We do not know where within that interval the animals were actually located. This is explained in the text.

*For figures 4-10 it might be helpful to have a title to the overall figure, describing which category each group of species belongs to and/or what type of data it represents, simply for quick reference, as they are all very similar. In the published manuscript the caption will be with the figure so this may not be necessary, but because they are all so similar, there might be a way to set them apart.*

We will add titles to the figures.
We added titles to the figures.

*Table 3 appears unfinished. I think the first column with data names should be reformatted and split into multiple columns for the different metadata and using actual words and not abbreviations (e.g. "D" and "N")*

We will break up the first column into several component columns to be more easily understood. We will look into gray shading for the night value rows (if allowed by the journal). We are trying to make this table fit into 1 page for the journal.
We revised the first few columns to make the table easier to understand.

*The discussion of oil presence in E. inermis is interesting, but the text presents data not shown in any figures, and it would be better to note that this data is not shown or show it. In particular the inclusion of the percentages of individuals with oil in them should be cited in some way, and perhaps an indication of how many individuals were observed. If this is not a fully quantified number (e.g. if it is anecdotal) it may still be important and worthy of inclusion but it needs some documentation.*

We will include exact numbers of individuals examined in the text. This was quantitatively done only for the SJ07 cruise.
We added more about the oil measurements in the Methods (lines 191 – 194) and Results sections (sect. 3.2.2) and re-organized that section. We added numbers of individuals observed.

**Reviewer 2.**

*The strength of the manuscript is also a weakness: it contains quite a lot of high-quality data (which is valuable to the scientific community) but as is, it does not well concatenate information, and a statistical analysis is entirely lacking. Given that mean T, S, O2, Chl-a values are available for each sample, it should be attempted to*

*tease out the main environmental drivers regulating the vertical distribution at day and night, and to present a physiological niche in which the respective species is to be found.*

A statistical model as described above is beyond the scope of this paper, which focuses on a descriptive presentation of individual species distributions.  The data are available to a modeler in the future who would like to do this sort of analysis.  Supplementary table S1 presents abundance data for each net and is available in digital spreadsheet form at the URI Digital Commons (presently available only for reviewers but will be made open access when the paper is published).  MOCNESS event logs are in the BCO-DMO database.  MOCNESS hydrographic data are available by request from the first author.  As described above, we believe that hydrographic ranges for a net, rather than means, are more pertinent to individual species distributions and better represent the uncertainty inherent in MOCNESS sampling and the nuances of the distributions.
DOI numbers for these tables and datasets are provided in the Data Availability section.

Also, in the absence of physiological data for most species, it is not possible to tease out the causes of the vertical distributions, day or night.   Depending on species-specific physiology, aerobic scope (a key determinant of biogeography) may be variably oxygen-limited, cold-limited, heat-limited, or irrelevant.

*Since the metabolic implications are discussed in some detail, I was wondering why environmental oxygen concentrations, rather than pO2, are reported throughout the paper. It would be much easier for the reader to understand the constraints, in particular for those species where pcrit data are available (consider extrapolation as a function of temperature).*

Oxygen concentration is included in the datastream of the MOCNESS and thus is readily available and is the common parameter used in most hydrographic literature.  We will provide PO2 for comparison parenthetically at a specified temperature.

Only a few of these species have been kept alive in a lab for Pcrit measurements.  For those species for which Pcrit has been measured at more than one temperature, we can discuss constraints but are not comfortable extrapolating too far given the inverse temperature effect in *L. hulsemannae* and the limited temp range (5 to 8°C).  We have Pcrit data for only 3 of these species.  Among them, the effect of temperature is normal, inverse and zero.  We will add more about those species to the metabolic discussion.
We revised the metabolic discussion section 4.6

*Fig. 1: This map does not reveal much oceanographic information to the reader. Consider including e.g. oxygen contours or average annual surface productivity. Lat/Lon grid should rather be equally spaced (I understand that the goal was to add the approximate lan/lon values for the sampling stations, but the exact values are given in the metadata table, and linear axes make it easier for the reader to visually grasp area size and distances.*

Fig. 1 is intended to be a simple schematic showing the geographic locations of the stations.  We will change the axis labels to even lat lon values.  Contours of environmental parameters are not pertinent.  We did not conduct regional surveys, and this work was done on 4 separate cruises over a time period of 10 years.  We provide references to the literature on the basic oceanography of the Eastern Tropical Pacific.
We revised axis labels.

Fig. 2. See combined Reviewer 1 and 2 comments above.

*Figure 4-10: These are way too many figures, they are difficult to read, and they don't convey as much information as they could. Sometimes the panels are organized in a confusing way (e.g. plots from the same area are not next to each other). I suggest to move the majority of these into a supplement, and only keep more integrative figures in the manuscript (which could be, e.g., scatter plots of multivariate analyses or histograms of abundance distribution against oxygen and/or temperature rather than single station profiles).*

Plots from the same area are arrayed vertically in columns, not horizontally (see station labels at the top of each column).

As noted elsewhere, the focus of this paper is on the descriptive presentation of how a number of copepod species respond to OMZ extent and the nuances of their responses to both the oxygen values and the shape of the oxygen profiles.  Thus, these figures are the crux of the paper.  A scatterplot based on means would not adequately illuminate the many possibilities of how individual species respond to both large and subtle changes in these profiles at particular times and places.  This paper provides a comprehensive abundance and distributional framework that will hopefully inspire further analyses in the future.  As noted above for Reviewer 1, we will develop a summary schematic diagram to highlight basic conclusions.  See the new schematic diagram Fig. 11.

*As for the stacked bar charts, I recommend variable bar width so that the bar covers the entire depth stratum sampled as there are no "gaps" between nets (this way, also the colors are more visible). Bar area then is proportional to integrated abundance in the respective depth layer. Day/Night plots of the same station should be scaled the same, and might be mirrored against each other to save space and facilitate comparison.*

We will work on these figures.
We made the bars thicker and improved labelling.  As explained in the caption, they are in sampling order by depth.

*I have added some additional, specific comments to a marked-up version of the pdf.*

We note the textual suggestions recommended by the reviewer.
We made most of the suggested textual changes.

[revised manuscript text omitted]

        **Supplementary Material**

        **Figure S1:  Oxygen profile variability.**  All oxygen profiles at each station are shown for the vertically-stratified MOCNESS tows used for zooplankton distributions discussed in this paper.  These are from the upcast portion of the tow during zooplankton
910     sampling.  In some cases, portions of tows were not available because of sensor issues.

        **Table S1:**  Details for each net in a tow including depth range and volume filtered, followed by abundances in each net (number $(1000 \text{ m})^{-3}$) for each of the species considered in this paper (Wishner et al., 2019, https://digitalcommons.uri.edu/gso-data/1/). Species names are shortened; full names are in Table 2.

[revised manuscript text omitted]

The following is a large rotated data table. It is organized into six measurement sections (DMA Net ID; Max Abund (#/1000m3); DMA (m); DMA Ox (µM); DMA Temp (°C); Water Col Abund (#/m2)). Within each section the eight Cruise / Day-or-Night samples are: CRD07 (D), CRD07 (N), TB07 (D), TB08 (D), OC (D), OC (N), SKQ (D), SKQ (N).

**DMA Net ID**

| Species | CRD07 D | CRD07 N | TB07 D | TB08 D | OC D | OC N | SKQ D | SKQ N |
|---|---|---|---|---|---|---|---|---|
| Disseta palumbii | 6231 | 6198 | x | 7135 | 7096 | 7278 | 7162 | 7162 |
| Euaugaptil magnus | x | x | x | 7105 | 7093 | 7273 | 7162 | 7162 |
| Euaugaptil nodifrons | x | x | x | 7081 | 7091 | 7221 | 7162 | 7162 |
| Eucalanus californicus | none | none | x | 7107 | 7071 | 7276 | 7162 | none |
| Eucalanus spinifer | none | x | x | 7108 | 7071 | 7276 | 7184 | 7165 |
| Gaetanus kruppi | 6232 | 6198 | x | 7081 | 7095 | 7272 | 7272 | 7162 |
| Gaetanus pseudodaiaf | none | 6193 | x | 7102 | 7092 | 7272 | 7272 | 7161 |
| Heterostylt long/long | 6237 | 6173 | x | 7111 | 7077 | 7194 | 7207 | 7207 |
| Lophothrix frontalis | x | x | x | 7111 | 7077 | 7186 | 7213 | 7213 |
| Luciculia hulsemann | 6236 | 6173 | 6072 | 7108 | 7098 | 7254 | 7201 | 7201 |
| Luciculia ovalis | none | 6191 | x | 6311 | 7104 | 7094 | none | none |
| Metridia brevicauda | 6165 | 6156 | 6316 | 7117 | 7071 | 7196 | 7196 | 7208 |
| Metridia princeps | 6233 | 6194 | x | 7111 | 7073 | 7255 | 7166 | 7166 |
| Pareuchta californica | x | x | x | 7081 | 7073 | 7274 | 7162 | 7162 |
| Scaphocal magnus | x | x | x | 7112 | 7071 | 7278 | 7162 | 7162 |
| Eucalanus inermis | 6165 | 6155 | 6077 | 6316 | 7112 | 7071 | 7193 | 7203 |
| Pleuroma abdominali | 6166 | 6217 | none | none | 7084 | 7077 | 7221 | 7218 |
| Pleuroma johnsoni | 6163 | 6173 | 6072 | 6296 | none | 7077 | 7196 | 7217 |
| Pleuroma quadnungu | none | none | none | none | 7083 | 7077 | 7221 | 7211 |
| Haloptilus longicornis | 6181 | 6212 | | 6263 | 7084 | 7074 | 7257 | 7217 |
| Luciculia flavicornis | 6181 | 6217 | | 6264 | 7088 | 7078 | 7257 | 7169 |
| Mecynocer clausi | 6187 | 6216 | x | 6265 | 7088 | 7078 | 7227 | 7218 |
| Subeucala subtenuis | 6187 | 6218 | 6078 | 6264 | 7088 | none | 7227 | 7217 |

**Max Abund (#/1000m3)**

| Species | CRD07 D | CRD07 N | TB07 D | TB08 D | OC D | OC N | SKQ D | SKQ N |
|---|---|---|---|---|---|---|---|---|
| Disseta palumbii | 23 | 25 | x | x | 40.6 | 24.8 | 53.8 | 22.1 |
| Euaugaptil magnus | x | x | x | x | 2.7 | 1 | 3.7 | 1.1 |
| Euaugaptil nodifrons | x | x | x | x | 1.3 | 0.8 | 5.3 | 2.2 |
| Eucalanus californicus | none | none | x | x | 3.1 | 11.5 | 1.9 | 0 |
| Eucalanus spinifer | none | none | x | x | 6.4 | 11.5 | 4.6 | 3 |
| Gaetanus kruppi | 42 | 25 | x | x | 11.3 | 2.2 | 2.7 | 1.1 |
| Gaetanus pseudodaiaf | 0 | 21 | x | 1 | 0.8 | 8.9 | 1.9 | |
| Heterostylt long/long | 1004 | 1151 | x | 1040.8 | 10.9 | 46.4 | 20.3 | 74.7 |
| Lophothrix frontalis | x | x | x | 63.7 | 6.8 | 44.4 | 16.9 | |
| Luciculia hulsemann | 997.3 | 64 | 288 | 780.6 | 82.8 | 91.9 | 81.9 | 65.2 |
| Luciculia ovalis | 0 | 75 | x | 63.2 | 1.2 | 1.9 | none | none |
| Metridia brevicauda | 15958 | 8542 | x | 1962.2 | 10.5 | 26.8 | 3.1 | 4.7 |
| Metridia princeps | 49 | 17 | x | 13.1 | 8.1 | 17.6 | 3.6 | |
| Pareuchta californica | x | x | x | 3.8 | 6.5 | 4.9 | 1.1 | |
| Scaphocal magnus | x | x | x | 14.5 | 7.7 | 17.9 | 7.7 | |
| Eucalanus inermis | 33912 | 16539 | 5255.4 | 4905.5 | 41.6 | 69 | 226.8 | 192.6 |
| Pleuroma abdominali | 3251 | 13386 | none | none | 80.2 | 31 | 288.2 | 382.1 |
| Pleuroma johnsoni | 40 | 64 | 1234.3 | 3263.6 | none | none | 41.4 | 30.7 |
| Pleuroma quadnungu | none | none | none | none | 38.3 | 201.3 | 202.8 | 38.5 |
| Haloptilus longicornis | 2210 | 1515 | x | 508.1 | 26.7 | 83.2 | 6.8 | 46.1 |
| Luciculia flavicornis | 943 | 6693 | x | 13757.8 | 144.8 | 120 | 10.6 | 4.1 |
| Mecynocer clausi | 11216 | 8092 | x | 15215.5 | 144.8 | 48 | 10.6 | 15.9 |
| Subeucala subtenuis | 33647 | 21864 | 1989.1 | 33336.1 | 24.1 | none | 5.3 | 15.4 |

**DMA (m)**

| Species | CRD07 D | CRD07 N | TB07 D | TB08 D | OC D | OC N | SKQ D | SKQ N |
|---|---|---|---|---|---|---|---|---|
| Disseta palumbii | 732-773 | 700-750 | x | 675-700 | 600-650 | 611-633 | 700-800 | |
| Euaugaptil magnus | x | x | x | 650-700 | 750-800 | 725-778 | 700-800 | |
| Euaugaptil nodifrons | x | x | x | 400-450 | 850-1000 | 300-350 | 700-800 | |
| Eucalanus californicus | none | none | x | 550-600 | 400-450 | 652-681 | none | |
| Eucalanus spinifer | none | none | x | 450-550 | 400-450 | 475-500 | 400-500 | |
| Gaetanus kruppi | 700-733 | 700-750 | x | 400-450 | 650-700 | 778-813 | 700-800 | |
| Gaetanus pseudodaiaf | none | 950-1000 | x | 800-850 | 800-850 | 778-813 | 800-1000 | |
| Heterostylt long/long | 575-600 | 550-747 | x | 800-825 | 200-250 | 100-150 | 475-500 | 400-425 |
| Lophothrix frontalis | x | x | x | 460-468 | 100-150 | 425-450 | 200-250 | |
| Luciculia hulsemann | 600-625 | 550-750 | 750-900 | 800-825 | 450-550 | 450-550 | 500-600 | 600-650 |
| Luciculia ovalis | none | 1100-1200 | 900-1000 | 700-750 | 700-750 | none | none | |
| Metridia brevicauda | 300-350 | 250-300 | 775-800 | 367-385 | 400-450 | 425-450 | 350-400 | |
| Metridia princeps | 675-700 | 900-950 | x | 460-468 | 300-350 | 400-500 | 300-400 | |
| Pareuchta californica | x | x | x | 400-450 | 300-350 | 704-725 | 700-800 | |
| Scaphocal magnus | x | x | x | 460-463 | 400-450 | 611-633 | 700-800 | |
| Eucalanus inermis | 300-350 | 300-350 | 20-80 | 775-800 | 460-463 | 400-450 | 500-550 | 500-550 |
| Pleuroma abdominali | 250-300 | 20-30 | none | 250-300 | 100-150 | 300-350 | 50-75 | |
| Pleuroma johnsoni | 400-450 | 550-750 | 750-900 | 250-300 | none | 425-450 | 75-100 | |
| Pleuroma quadnungu | none | none | none | 300-350 | 100-150 | 300-350 | 300-350 | |
| Haloptilus longicornis | 100-150 | 80-100 | x | 60-80 | 250-300 | 250-300 | 200-300 | 75-100 |
| Luciculia flavicornis | 100-150 | 20-30 | x | 50-60 | 0-100 | 250-300 | 75-100 | 0-100 |
| Mecynocer clausi | 20-30 | 30-40 | x | 40-50 | 0-100 | 0-100 | 75-100 | 50-75 |
| Subeucala subtenuis | 20-30 | 0-20 | 0-20 | 50-60 | 0-100 | none | 75-100 | 75-100 |

**DMA Ox (µM)**

| Species | CRD07 D | CRD07 N | TB07 D | TB08 D | OC D | OC N | SKQ D | SKQ N |
|---|---|---|---|---|---|---|---|---|
| Disseta palumbii | 7.5-9.7 | 7.8-8.4 | x | 5.8-6.3 | 4.4-5.8 | 4.0-4.8 | 3.2-5.3 | |
| Euaugaptil magnus | x | x | x | 5.4-6.6 | 7.4-8.9 | 4.2-7.4 | 3.2-5.3 | |
| Euaugaptil nodifrons | x | x | x | 6.5-8.2 | 10.4-15.4 | 6.8-20.1 | 3.2-5.3 | |
| Eucalanus californicus | none | none | x | 4.3-4.8 | 4.1-5.4 | 3.2-5.9 | none | |
| Eucalanus spinifer | none | none | x | 4.4-5.3 | 4.1-5.4 | 4.2-6.1 | 3.4-7.8 | |
| Gaetanus kruppi | 7.4-8.3 | 7.8-8.4 | x | 6.5-8.2 | 5.6-6.8 | 6.6-9.1 | 3.2-5.3 | |
| Gaetanus pseudodaiaf | none | 18.4-24.8 | x | 8.9-10.6 | 8.9-10.7 | 6.6-9.1 | 4.7-14.8 | |
| Heterostylt long/long | 1.3-2.0 | 1.1-7.6 | x | 1.84-2.41 | 25.6-69.3 | 97.3-195 | 1.0-1.7 | 1.0-1.9 |
| Lophothrix frontalis | x | x | x | 4.6-5.0 | 97.3-195 | 5.0-7.4 | 39.5-65.5 | |
| Luciculia hulsemann | 1.7-5.9 | 1.1-7.6 | 1.75-7.20 | 1.84-2.41 | 4.4-5.3 | 3.6-4.6 | 3.0-6.8 | 1.5-3.3 |
| Luciculia ovalis | none | 28.9-35.2 | x | 5.7-6.9 | 6.4-7.4 | 6.4-7.6 | none | none |
| Metridia brevicauda | 1.0-5.7 | 1.3-1.6 | 1.4-1.9 | 7.1-10.2 | 4.1-5.4 | 1.2-5.7 | 1.3-4.9 | |
| Metridia princeps | 7.1-8.0 | 14.6-18.7 | x | 4.6-5.0 | 11.7-17.8 | 6.8-11.3 | 1.3-7.1 | |
| Pareuchta californica | x | x | x | 6.5-8.2 | 11.7-17.8 | 4.5-6.1 | 3.2-5.3 | |
| Scaphocal magnus | x | x | x | 4.4-5.0 | 4.1-5.4 | 4.0-4.8 | 3.2-5.3 | |
| Eucalanus inermis | 1.0-5.7 | 1.2-1.6 | 3.0-200 | 1.4-1.9 | 4.4-5.0 | 4.1-5.4 | 1.2-2.2 | 1.0-1.9 |
| Pleuroma abdominali | 3.4-8.6 | NA | none | 15.7-32.0 | 97.3-195 | 6.8-20.1 | 202-205 | |
| Pleuroma johnsoni | 1.0-1.7 | 1.1-7.6 | 1.8-7.2 | 1.2-1.4 | none | 1.2-5.7 | 203-215 | |
| Pleuroma quadnungu | none | none | none | 12.7-17.4 | 6.8-20.1 | 6.3-21.5 | | |
| Haloptilus longicornis | 13.1-21.6 | 16.7-22 | x | 1.3-2.0 | 15.7-32.0 | 17.3-27.9 | 15.6-61.2 | 203-215 |
| Luciculia flavicornis | 13.1-21.6 | NA | x | 3.9-35.7 | 203-231 | 189-228 | 207-214 | 170-210 |
| Mecynocer clausi | NA | 28.0-28.0 | x | 39.3-74.5 | 203-231 | 189-228 | 207-214 | 202-205 |
| Subeucala subtenuis | NA | NA | 196-200 | 3.9-35.7 | 203-231 | 207-214 | 203-215 | |

**DMA Temp (°C)**

| Species | CRD07 D | CRD07 N | TB07 D | TB08 D | OC D | OC N | SKQ D | SKQ N |
|---|---|---|---|---|---|---|---|---|
| Disseta palumbii | 5.4-5.7 | 5.7-6.1 | x | 5.7-5.9 | 6.0-6.4 | 6.2-6.4 | 5.4-5.9 | |
| Euaugaptil magnus | x | x | x | 5.7-6.0 | 5.2-5.4 | 5.3-5.7 | 5.4-5.9 | |
| Euaugaptil nodifrons | x | x | x | 7.6-8.1 | 4.4-4.9 | 9.6-9.9 | 5.4-5.9 | |
| Eucalanus californicus | none | none | x | 6.3-6.6 | 7.9-8.3 | 5.8-6.2 | none | |
| Eucalanus spinifer | none | none | x | 6.6-7.7 | 7.9-8.3 | 7.7-7.9 | 7.6-9.0 | |
| Gaetanus kruppi | 5.7-5.9 | 5.7-6.1 | x | 7.6-8.1 | 5.6-6.0 | 5.1-5.2 | 5.4-5.9 | |
| Gaetanus pseudodaiaf | none | 4.5-4.7 | x | 4.9-5.1 | 4.9-5.2 | 5.1-5.2 | 4.4-5.4 | |
| Heterostylt long/long | 6.5-6.6 | 5.8-7.4 | x | 5.5-5.6 | 10.4-11.0 | 11.7-15.1 | 7.6-8.1 | 8.7-9.0 |
| Lophothrix frontalis | x | x | x | 7.5-7.7 | 11.7-15.1 | 8.1-8.5 | 10.5-11.5 | |
| Luciculia hulsemann | 6.3-6.5 | 5.8-7.4 | 5.21-5.90 | 5.5-5.6 | 6.6-7.7 | 6.8-7.7 | 6.8-7.3 | 6.2-6.7 |
| Luciculia ovalis | none | 3.7-4.2 | x | 4.7-5.1 | 5.4-5.7 | 5.4-5.7 | none | none |
| Metridia brevicauda | 10.0-10.8 | 10.8-11.6 | 5.6-5.8 | 7.9-8.3 | 8.3-8.6 | 9.0-9.4 | | |
| Metridia princeps | 5.9-6.0 | 4.7-4.9 | x | 7.5-7.7 | 8.8-9.4 | 7.3-8.5 | 9.0-10.8 | |
| Pareuchta californica | x | x | x | 7.6-8.1 | 8.8-9.4 | 5.6-5.7 | 5.4-5.9 | |
| Scaphocal magnus | x | x | x | 7.7-7.8 | 7.9-8.3 | 6.2-6.4 | 5.4-5.9 | |
| Eucalanus inermis | 10.0-10.8 | 10.1-10.8 | 13.5-27.4 | 5.6-5.8 | 7.7-7.8 | 7.9-8.3 | 7.0-7.6 | 7.2-7.8 |
| Pleuroma abdominali | 10.8-11.6 | 17.8-26.7 | none | 9.6-10.4 | 11.7-15.1 | 9.5-9.9 | 22.6-22.7 | |
| Pleuroma johnsoni | 8.2-9.1 | 5.8-7.4 | 5.2-5.9 | 10.8-11.3 | none | 8.3-8.6 | 20.3-22.7 | |
| Pleuroma quadnungu | none | none | none | 12.7-17.1 | 8.8-9.6 | 11.7-15.1 | 9.5-9.9 | 9.6-9.9 |
| Haloptilus longicornis | 12.7-13.4 | 13.8-14 | x | 14.4-16.9 | 9.6-10.4 | 9.4-10.2 | 9.9-11.2 | 20.3-22.7 |
| Luciculia flavicornis | 12.7-13.4 | 17.8-26.7 | x | 17.2-20.8 | 15.8-21.7 | 15.1-21.7 | 19.3-21.4 | 19.8-22.7 |
| Mecynocer clausi | 15.6-19.2 | 15.6-17.3 | x | 21.1-23.9 | 15.8-21.7 | 15.1-21.7 | 19.3-21.4 | 22.6-22.7 |
| Subeucala subtenuis | 15.6-19.2 | 26.8-26.8 | 27.5-27.7 | 17.2-20.8 | 15.8-21.7 | 19.3-21.4 | 20.3-22.7 | |

**Water Col Abund (#/m2)**

| Species | CRD07 D | CRD07 N | TB07 D | TB08 D | OC D | OC N | SKQ D | SKQ N |
|---|---|---|---|---|---|---|---|---|
| Disseta palumbii | 0.5 | 2.7 | x | x | 7 | 6 | 4.5 | 4.8 |
| Euaugaptil magnus | x | x | x | x | 0.7 | 0.1 | 0.8 | 0.1 |
| Euaugaptil nodifrons | x | x | x | x | 0.1 | 0.2 | 0.5 | 0.2 |
| Eucalanus californicus | 0 | 0 | x | x | 0.3 | 1 | 0.1 | 0 |
| Eucalanus spinifer | 0 | 0 | x | x | 1.2 | 1 | 0 | 0.6 |
| Gaetanus kruppi | 2.1 | 1.8 | x | x | 0.8 | 0.2 | 0.5 | 0.1 |
| Gaetanus pseudodaiaf | 0 | 2.5 | x | x | 0.2 | 0.1 | 1.1 | 0.4 |
| Heterostylt long/long | 78.5 | 230 | x | 60 | 1.4 | 4.8 | 0.4 | 2.3 |
| Lophothrix frontalis | x | x | x | 63.7 | 0.9 | 0.7 | 2.8 | 1.1 |
| Luciculia hulsemann | 28.7 | 9.6 | 43.2 | 39.6 | 10.6 | 10.6 | 15.6 | 7.1 |
| Luciculia ovalis | 0 | 7.5 | x | 6.3 | 0.2 | 0.2 | none | none |
| Metridia brevicauda | 984.3 | 755.8 | 107.8 | 0.6 | 2.3 | 0.2 | 1.1 | |
| Metridia princeps | 1.2 | 1.4 | x | 1.3 | 1.2 | 4.1 | 1 | |
| Pareuchta californica | x | x | x | 0.5 | 0.9 | 0.4 | 0.3 | |
| Scaphocal magnus | x | x | x | 1.2 | 1.1 | 4 | 2.8 | |
| Eucalanus inermis | 2182.5 | 1345.5 | 348.6 | 246 | 1.7 | 6.5 | 3.2 | 29.9 |
| Pleuroma abdominali | 217.5 | 353.7 | 0 | 6.5 | 2.2 | 2.5 | 7.2 | |
| Pleuroma johnsoni | 2.5 | 10.9 | 302.7 | 435.4 | none | 1.4 | 0.2 | |
| Pleuroma quadnungu | 0 | 0 | 0 | 3.7 | 13.2 | 9 | 5.5 | |
| Haloptilus longicornis | 302.4 | 122.5 | x | 13.3 | 1.8 | 12.9 | 1.6 | 0.7 |
| Luciculia flavicornis | 62.1 | 179.3 | x | 171.1 | 16.1 | 12.8 | 0.8 | 0.5 |
| Mecynocer clausi | 117.7 | 125.1 | x | 254.3 | 14.5 | 4.8 | 0.2 | 0.2 |
| Subeucala subtenuis | 594.9 | 608 | 118.9 | 490.9 | 2.4 | 0 | 0.3 | 0.1 |

---

## Author Response (AR3)

**Response to the Editor (3/31/20)**

**Comments to the Author:**

The splitting of the too-large Table 3 into two tables (now Tables 3 and 4) is a reasonable approach for reducing the length and complexity of the original Table 3. However, the text needs additional updating to reflect that there are now two tables and the text should provide more guidance to the reader where to find the specific information in the Tables that are cited to support the more general statements in the results. For clarity, I extracted two examples where additional citing in the text of numbers that appear in tables and where the citing of a table seems to be missing.

Example 1: E. inermis diapausing layers were located at extremely low oxygen, 1.0-5.7 $\mu$M (except for the shallow layer at TB in 2008 where the net probably sampled across zones) (Table 3).

I think you need to also refer to Table 4 so the reader knows why you cite 1.0-5.7 uM.

Example 2: Oxygen where M. brevicauda was most abundant was 1.0-5.7 $\mu$M with a temperature of 10.0 – 10.8°C (Table 3) in the UO of CRD in 2007 (300-350 m, maximum adult abundance in this net of 15,958 (1000 m)-3 ), ...

I believe the maximum abundance is in Table 4, but Table 4 is not cited in this example.

I suggest you go through the results that rely on Tables 3 and 4, and expand the text a bit to cite the numbers in the Tables (or specific rows with columns) as the values are used to support statements in the text. Basically, please modify and add to the text information that carefully leads the reader through the use of Tables 3 and 4. This does not affect your results but rather makes it easy for the reader to understand the basis of the results statements.

**Response:**

We have added more detailed references in the text to Tables 3 and 4 and also added a few more details in the table captions to make it easier to find the species. It would be too voluminous to add specific line and row numbers in the text for every reference to those tables.

I thank the editor for a fast turnaround, and I would appreciate a similar fast response so that the paper can be finally accepted and the publication process can proceed. I am bundling several sources of funding for the publication charges (some from the university), and some of those will be expiring very soon. I will need to pay additional charges personally if those funding sources expire. The journal staff has said that I cannot receive an invoice in advance from the journal before the publication process. So I am eager to get publication underway as soon as the paper is accepted.

[revised manuscript text omitted]
 | 7208 | 7166 | 7162 | 7162 | 7203 | 7218 | 7217 | 7211 | 7217 | 7169 | 7218 | 7217 |

---

## Author Response (AR4)

**Response to reviewers:  3 Feb 2020**
We appreciate the time and attention of the reviewers.  Below, we discuss our responses to their suggestions and what we intend to do.

**Reviewer 1**

*I think the paper would benefit from a better presentation of the data in tables and figures, and perhaps a summary diagram or two, as well as a summary table.*
*This would be a useful way to summarize the data: a table and/or diagram showing how each of the 4 categories (generally) responds to both habitats.*

We will prepare a summary diagram and consider a summary table.
We added schematic diagram Fig. 11.

*Fig. 2 Reviewer 1:  Figure 2 is somewhat confusing with different vertical and horizontal scales for the same data. The zoomed aspect is a good idea, but I think that the graphs should at least be rearranged, with shallower graphs above deeper one, and perhaps the deeper ones should have a depth minimum at the place where the shallower ones end. Also, a box around the region that is zoomed in could be usfeul, e.g. for the upper layer oxygen data, so that it looks more like an inset. For the TO plots on the bottom, I think a box around the region that is zoomed in will be helpful too.*

*Fig. 2 Reviewer 2: This is a very large and very busy figure, mainly due to the many different colors. First, I recommend using the mean profile instead of a chosen single profile for each station (maybe with shaded error, but this might overcrowd the graphs). Second, choose three colors that are the same or similar for the three regions. Try to make the figure fit into a page (lower panels are wider, legend is out of the figure). Oxygen profiles in these would be helpful. Plot area lines could be removed to make some space, but tick marks added because difficult to read with just one tick mark.*

We will add boxes to make these diagrams easier to understand.  We prefer to have the full water column profiles at the top for overview, and the zoomed upper water column region below, however.  We think that it is important to show individual station profiles, hence the different colors, one for each station. (What are the "3 regions" of Reviewer 2)?  Mean data would obscure the variability and nuances in distributions that we are trying to show (see also next comment below).  The lower panels in Fig 2 are wider (square) because they plot two hydrographic variables (salinity and temperature or oxygen and temperature), not depth. (We don't understand the "oxygen profiles in these" comment of Reviewer 2). Given the complexity of Fig. 2, we think that more tick marks would introduce too much clutter, but we will evaluate adding some.
We revised Fig 2 with additional labels and indicator boxes.

*I also wondered about the variability in a given station, because the CTD data is shown as one line from a representative cast, while the zooplankton data is an aggregate. Some comment or presentation of the variability, at least of O2 would be useful. As thin lines of the same color, or a shaded region around the O2 perhaps. This may not be possible and may make the graphs too busy, so even just a note in the text or some supplementary figures would help alleviate concerns about that variability.*

Understanding the variability of oxygen is a valid concern.  We will provide some supplementary figures in which all oxygen profiles from a particular station are shown to give a sense of variability.  At each station, there were multiple MOCNESS tows to different depths that we used to construct the full abundance profiles (see Table 1 and methods text).  However, adding all of the oxygen profiles to the abundance profile figures would really clutter them.
We added Supplementary Fig. S1 to show all the oxygen profiles at each station.

The point of the abundance figures is to highlight the depth of maximum abundance (DMA), i.e. where the species is most abundant, and the nuances of their distributions relative to the different shapes of the oxygen profiles. Depth, oxygen, and temperature ranges for the single net at the DMA are presented in Table 3. We believe that ranges, rather than means, are the most valid way to interpret their habitat because MOCNESS tow nets sample over a depth interval and consequently over a range of oxygen and temperature. We do not know where within that interval the animals were actually located. This is explained in the text.

*For figures 4-10 it might be helpful to have a title to the overall figure, describing which category each group of species belongs to and/or what type of data it represents, simply for quick reference, as they are all very similar. In the published manuscript the caption will be with the figure so this may not be necessary, but because they are all so similar, there might be a way to set them apart.*

We will add titles to the figures.
We added titles to the figures.

*Table 3 appears unfinished. I think the first column with data names should be reformatted and split into multiple columns for the different metadata and using actual words and not abbreviations (e.g. "D" and "N")*

We will break up the first column into several component columns to be more easily understood. We will look into gray shading for the night value rows (if allowed by the journal). We are trying to make this table fit into 1 page for the journal.
We revised the first few columns to make the table easier to understand.

*The discussion of oil presence in E. inermis is interesting, but the text presents data not shown in any figures, and it would be better to note that this data is not shown or show it. In particular the inclusion of the percentages of individuals with oil in them should be cited in some way, and perhaps an indication of how many individuals were observed. If this is not a fully quantified number (e.g. if it is anecdotal) it may still be important and worthy of inclusion but it needs some documentation.*

We will include exact numbers of individuals examined in the text. This was quantitatively done only for the SJ07 cruise.
We added more about the oil measurements in the Methods and Results sections (sect. 3.2.2) and re-organized that section. We added numbers of individuals observed.

**Reviewer 2.**

*The strength of the manuscript is also a weakness: it contains quite a lot of high-quality data (which is valuable to the scientific community) but as is, it does not well concatenate information, and a statistical analysis is entirely lacking. Given that mean T, S, O2, Chl-a values are available for each sample, it should be attempted to tease out the main environmental drivers regulating the vertical distribution at day and night, and to present a physiological niche in which the respective species is to be found.*

A statistical model as described above is beyond the scope of this paper, which focuses on a descriptive presentation of individual species distributions. The data are available to a modeler in the future who would like to do this sort of analysis. Supplementary table S1 presents abundance data for each net and is available in digital spreadsheet form at the URI Digital Commons (presently available only for reviewers but will be made open access when the paper is published). MOCNESS event logs are in the BCO-DMO database. MOCNESS hydrographic data are available by request from the first author. As described above, we believe that hydrographic ranges for a net, rather than means, are more pertinent to individual species distributions and better represent the uncertainty inherent in MOCNESS sampling and the nuances of the distributions.
DOI numbers for these tables and datasets are provided in the Data Availability section.

Also, in the absence of physiological data for most species, it is not possible to tease out the causes of the vertical distributions, day or night. Depending on species-specific physiology, aerobic scope (a key determinant of biogeography) may be variably oxygen-limited, cold-limited, heat-limited, or irrelevant.

*Since the metabolic implications are discussed in some detail, I was wondering why environmental oxygen concentrations, rather than pO2, are reported throughout the paper. It would be much easier for the reader to understand the constraints, in particular for those species where pcrit data are available (consider extrapolation as a function of temperature).*

Oxygen concentration is included in the datastream of the MOCNESS and thus is readily available and is the common parameter used in most hydrographic literature. We will provide PO2 for comparison parenthetically at a specified temperature.

Only a few of these species have been kept alive in a lab for Pcrit measurements. For those species for which Pcrit has been measured at more than one temperature, we can discuss constraints but are not comfortable extrapolating too far given the inverse temperature effect in *L. hulsemannae* and the limited temp range (5 to 8°C). We have Pcrit data for only 3 of these species. Among them, the effect of temperature is normal, inverse and zero. We will add more about those species to the metabolic discussion.

We revised the metabolic discussion section 4.6

*Fig. 1: This map does not reveal much oceanographic information to the reader. Consider including e.g. oxygen contours or average annual surface productivity. Lat/Lon grid should rather be equally spaced (I understand that the goal was to add the approximate lan/lon values for the sampling stations, but the exact values are given in the metadata table, and linear axes make it easier for the reader to visually grasp area size and distances.*

Fig. 1 is intended to be a simple schematic showing the geographic locations of the stations. We will change the axis labels to even lat lon values. Contours of environmental parameters are not pertinent. We did not conduct regional surveys, and this work was done on 4 separate cruises over a time period of 10 years. We provide references to the literature on the basic oceanography of the Eastern Tropical Pacific.

We revised axis labels.

Fig. 2. See combined Reviewer 1 and 2 comments above.

*Figure 4-10: These are way too many figures, they are difficult to read, and they don't convey as much information as they could. Sometimes the panels are organized in a confusing way (e.g. plots from the same area are not next to each other). I suggest to move the majority of these into a supplement, and only keep more integrative figures in the manuscript (which could be, e.g., scatter plots of multivariate analyses or histograms of abundance distribution against oxygen and/or temperature rather than single station profiles).*

Plots from the same area are arrayed vertically in columns, not horizontally (see station labels at the top of each column).

As noted elsewhere, the focus of this paper is on the descriptive presentation of how a number of copepod species respond to OMZ extent and the nuances of their responses to both the oxygen values and the shape of the oxygen profiles. Thus, these figures are the crux of the paper. A scatterplot based on means would not adequately illuminate the many possibilities of how individual species respond to both large and subtle changes in these profiles at particular times and places. This paper provides a comprehensive abundance and distributional framework that will hopefully inspire further analyses in the future. As noted above for Reviewer 1, we will develop a summary schematic diagram to highlight basic conclusions. See the new schematic diagram Fig. 11.

*As for the stacked bar charts, I recommend variable bar width so that the bar covers the entire depth stratum sampled as there are no "gaps" between nets (this way, also the colors are more visible). Bar area then is proportional to integrated abundance in the respective depth layer. Day/Night plots of the same station should be scaled the same,*

*and might be mirrored against each other to save space and facilitate comparison.*

We will work on these figures.
We made the bars thicker and improved labelling. As explained in the caption, they are in sampling order by depth.

*I have added some additional, specific comments to a marked-up version of the pdf.*

We note the textual suggestions recommended by the reviewer.
We made most of the suggested textual changes.

**Response to the Editor:  29 March 2020**

*The authors have sufficiently addressed the major comments of the reviewers. One remaining issue, and the reason I determined "Publish subject to minor revision", is the format of Table 3. The table seems quite important, as it is cited in the main text multiple times in support of statements of results. The present format of Table 3 makes it too difficult to understand and too difficult to find the specific values that support each results statement in the text. Perhaps breaking the table into multiple, smaller tables would be enough? Maybe extracting and summarizing the detailed data to highlight the key results in a new table or figure, and putting the individual cruise information in supplemental? Table 3 needs to be revised so that it is more readable and so that readers can more easily follow how the data in the table supports the results stated in the text. What I offered above are illustrative examples of changes to Table 3; I will leave the exact modifications to the authors.*

**Response**
We have split the former Table 3 into two tables.  The new Table 3 includes habitat parameters for the DMA net (depth, oxygen temperature ranges) for each species and cruise.  The new Table 4 includes pertinent abundance data for each species and cruise (maximum abundance as seen in the DMA net and water column abundance for the 0-1000 m depth interval).  Also, we re-formatted these tables for easier viewing.

**Response to the Editor:  31 March 2020**

*The splitting of the too-large Table 3 into two tables (now Tables 3 and 4) is a reasonable approach for reducing the length and complexity of the original Table 3. However, the text needs additional updating to reflect that there are now two tables and the text should provide more guidance to the reader where to find the specific information in the Tables that are cited to support the more general statements in the results. For clarity, I extracted two examples where additional citing in the text of numbers that appear in tables and where the citing of a table seems to be missing.*

*Example 1: E. inermis diapausing layers were located at extremely low oxygen, 1.0-5.7 μM (except for the shallow layer at TB in 2008 where the net probably sampled across zones) (Table 3).*

*I think you need to also refer to Table 4 so the reader knows why you cite 1.0-5.7 uM.*

*Example 2: Oxygen where M. brevicauda was most abundant was 1.0-5.7 μM with a temperature of 10.0 – 10.8°C (Table 3) in the UO of CRD in 2007 (300-350 m, maximum adult abundance in this net of 15,958 (1000 m)-3 ), ...*

*I believe the maximum abundance is in Table 4, but Table 4 is not cited in this example.*

*I suggest you go through the results that rely on Tables 3 and 4, and expand the text a bit to cite the numbers in the Tables (or specific rows with columns) as the values are used to support statements in the text. Basically, please modify and add to the text information that carefully leads the reader through the use of Tables 3 and 4. This does not affect your results but rather makes it easy for the reader to understand the basis of the results statements.*

**Response**
We have added more detailed references in the text to Tables 3 and 4 and also added a few more details in the table captions to make it easier to find the species.  It would be too voluminous to add specific line and row numbers in the text for every reference to those tables.

[revised manuscript text omitted]
 | 7208 | 7166 | 7162 | 7162 | 7203 | 7218 | 7217 | 7211 | 7217 | 7169 | 7218 | 7217 |